# Training Neural Networks from Scratch with Parallel Low-Rank Adapters

## Abstract

The scalability of deep learning applications is fundamentally constrained by compute, memory, and communication. While low-rank adaptation (LoRA) has reduced these costs for model fine-tuning, its application to model pre-training remain largely unexplored. This paper examines the extension of LoRA to model pre-training, identifying the constraints and limitations inherent to standard LoRA in the context of pre-training. We introduce *LoRA-the-Explorer* (LTE), a novel bi-level optimization algorithm, to facilitate parallel training of multiple low-rank heads across compute nodes, minimizing the necessity for frequent synchronization. Our methodology involves rigorous experimentation on vision transformers using ImageNet100, demonstrating that LTE is competitive with standard distributed training methodologies. Initial scalability tests on ImageNet1k show that LTE can match standard training performance by leveraging more training iterations.

## 1 Introduction

The escalating complexity and computational demands of state-of-the-art deep learning models present significant challenges, not just in terms of computational power cost but also in memory and communication bandwidth requirements. As these challenges exceed the capacities of consumer-grade GPUs, innovative solutions are imperative to further academic research. Low-rank adaptation (Hu et al., 2021, LoRA), has recently gained attention in this context, which uses low-rank parameterization of the model to reduce memory requirements for storing and communicating gradients/optimizer-state during training. With further innovations in parameter quantization (Dettmers et al., 2023), we now have tools that enables fine-tuning of large models in consumer-grade GPUs. However, these works are limited to fine-tuning, and tools to pre-train models from scratch are still absent to this day. Hence, the goal of this paper is to extend adaptation methods to model pre-training. Specifically, we posit the question: *Can neural networks be trained from scratch using low-rank adapters?*

Successfully addressing this question carries substantial implications, especially considering that common academic clusters often have slower cross-node training than single node training with gradient accumulation. Low-rank adapters effectively compresses the communication between these processors while preserving essential structural attributes for effective model training. Our investigation reveals that while vanilla LoRA underperforms in training a model from scratch, the use of parallel low-rank updates could bridge this performance gap. Our empirical analyses indicate that synchronization among multiple LoRA heads can occur sporadically, allowing computation to be re-budgeted towards training rather than synchronization. Our key findings are summarized as follows:

**Principal findings and contributions:**

- In Section 3, we establish the limitations inherent to the use of LoRA for model pre-training. We articulate the need for parallel updates and introduce our algorithmic approach LTE in Section 3.1 and Section 3.2.
- Section 4 shows competitive performance of LTE, even with infrequent synchronization.
- We provide rigorous empirical analysis and ablation study in Section 4.1, Section 4.2, and Section 4.3.
- In Section 4.5, we conduct a resource utilization comparison with standard distributed data parallel (DDP) training on 8 GPUs. Our findings indicate that, although our method extends the training duration by 40%, we can fit models that are 3× bigger with roughly half the bandwidth.
- Section 5 discusses related work. Supporting experiments and training details are detailed in Appendix A.

Our work presents a scientific investigation into using LoRA to pre-train a neural network from scratch. Our proposed method offers a proof of concept that, by trading off memory and computational resources, we can train large models from scratch using only low-rank adapters; a previously unexplored avenue. This research lays the groundwork for future work in scaling up large model training on low-memory GPU devices with slow interconnect speeds." Code to replicate our results will be open to public after the reviewing process.

## 2 PRELIMINARIES

**Notation**  We use $x$ to denote a scalar, $\mathbf{x}$ a vector, $X$ a matrix, $\mathcal{X}$ a distribution or a set, $f(\cdot)$ a function and $F(\cdot)$ a composition of functions. We use $l(\cdot)$ to denote a per-sample loss, and denote the average loss of a network $F(\cdot)$ equipped with parameters $\mathbf{W}$ as $\mathcal{L}(\mathbf{X}, \mathbf{Y}, \mathcal{F}, \mathbf{W}) = \frac{1}{|\mathbf{X}|} \sum_{(\mathbf{x}_i, \mathbf{y}_i) \sim (\mathbf{X}, \mathbf{Y})} l(F(\mathbf{x}_i; \mathbf{W}), \mathbf{y}_i)$. We will often write just $\mathcal{L}(\mathbf{X}, \mathbf{Y})$ for brevity.

### 2.1 LOW-RANK ADAPTER: PARAMETER-EFFICIENT ADAPTERS

Adapters serve as trainable functions that modify existing layers in a neural network. They facilitate efficient fine-tuning of large-scale models by minimizing both the computational cost of gradient calculations and the memory requirements for optimization. The focus of this work is on the *low-rank adapter* (Hu et al., 2021, LoRA), a subclass of linear adapters. The linearity of LoRA allows for the trained parameters to be integrated into the existing weights post-training, thereby maintaing the original inference cost. LoRA is employed in fine-tuning transformers, accounting for less than $10\%$ of the total model parameters (and even as low as $0.5\%$).

**Low-rank adapter (LoRA)**  We are given a linear layer $f : \mathbb{R}^n \to \mathbb{R}^m$, parameterized by weights $\mathbf{W} \in \mathbb{R}^{m \times n}$ that operates on input $\mathbf{x} \in \mathbb{R}^n$. LoRA parameterization uses low-rank matrices $\mathbf{B} \in \mathbb{R}^{m \times r}$, $\mathbf{A} \in \mathbb{R}^{r \times n}$ and a fixed scalar $s \in \mathbb{R}$, where the rank $r$ is chosen such that $r \ll \min(m, n)$.

$$f_{\mathsf{lora}}(\mathbf{x}) = \mathbf{W}\mathbf{x} + s\mathbf{B}\mathbf{A}\mathbf{x} \tag{1}$$

Although the forward pass incurs a minor computational overhead, the significance of LoRA parameterization pertains to the optimizer memory footprint. Optimizers such as AdamW (Kingma & Ba, 2014; Loshchilov & Hutter, 2017) typically maintain two states for each parameter, resulting in memory consumption that is twice the size of the trainable parameters. In the case of LoRA parameterization, the optimizer memory scales with the combined sizes of $\mathbf{A}$ and $\mathbf{B}$. This results in significant memory savings when the memory cost of LoRA $\mathcal{O}(r(m+n))$ is less than the memory cost of the model $\mathcal{O}(mn)$. Moreover, Hu et al. (2021) and Dettmers et al. (2023) demonstrated memory savings by storing $\mathbf{W}$ in low-precision while keeping the trainable parameters $\mathbf{A}$ and $\mathbf{B}$ in high-precision. These works have catalyzed the development of several repositories (Wang, 2023; Dettmers et al., 2023; Dettmers, 2023; huggingface, 2023), thereby enabling fine-tuning of models with billions of parameters on consumer-grade GPUs.

## 3 METHOD

Although low-rank adapters (LoRAs) have proven to be effective for fine-tuning tasks, their limitations become apparent when training models from scratch. As evidenced in Figure 1, models parameterized with LoRA demonstrate inferior performance compared to models trained using standard optimization. This performance gap can be attributed to the inherent rank constraint in LoRA. Specifically, for parameter $\mathbf{W} \in \mathbb{R}^{m \times n}$, LoRA is fundamentally incapable of recovering weights that exceed the rank $r < \min(m, n)$. Of course, there are exceptions in which, by happenstance, a solution exists within a low-rank proximity of the initialization. However, in Appendix B, we observed the rank of the gradient tends to increase throughout training, creating a necessity for high-rank updates.

To understand the conditions required to pre-train a model with LoRA, we first identify a specific scenario where standard training performance can be recovered using LoRA. This serves as a guiding principle for developing an algorithm that retains the computational efficiency intrinsic to LoRA.

### 3.1 MOTIVATION: MULTI-HEAD MERGING PERSPECTIVE

This section provides intuition on why training many LoRA heads in parallel with periodic merging is able to approximate standard full-rank pre-training.

As demonstrated in Figure 1, elevating the rank $r$ of the LoRA to be the same as the rank $\min(m, n)$ of the weight matrix $\mathbf{W} \in \mathbb{R}^{m \times n}$ is sufficient to replicate standard pre-training performance, albeit with different inherent dynamics as detailed in Appendix B.2. However, such an approach compromises the memory efficiency of low-rank adapters. Given that memory constraints often serve as significant bottlenecks in model training, we ask: *Can equivalent performance be achieved by concurrently training models with low-rank adapters?*

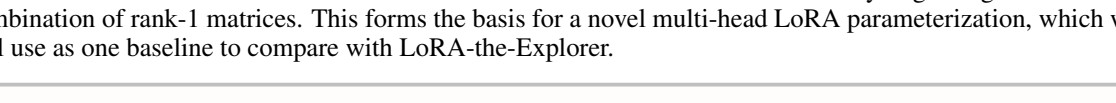

Figure 1: **Full-model vs. LoRA pre-training:** ViT-S trained on ImageNet100 using with and without LoRA. Low-rank LoRA uses rank $r = 64$ and full-rank LoRA uses rank $r = \min(m, n)$ set to the dimension of the original weight $\mathbf{W} \in \mathbb{R}^{m \times n}$. Increasing $r$ suffices to match standard training performance.

Given a matrix of the form $\mathbf{BA} \in \mathbb{R}^{d_1 \times d_2}$ with $\mathbf{B} \in \mathbb{R}^{d_1 \times d}$ and $\mathbf{A} \in \mathbb{R}^{d \times d_2}$, it is possible to represent it as the sum of two lower-rank matrices: $\mathbf{B}_1 \mathbf{A}_1 + \mathbf{B}_2 \mathbf{A}_2$. To demonstrate this, let $\mathbf{b}_i$ and $\mathbf{a}_i$ be the column vectors of $\mathbf{B}$ and $\mathbf{A}$ respectively. One can then construct $\mathbf{B}_1 = [\mathbf{b}_1, \ldots, \mathbf{b}_{[d/2]}]$, $\mathbf{B}_2 = [\mathbf{b}_{[d/2]}, \ldots, \mathbf{b}_d]$, and $\mathbf{A}_1 = [\mathbf{a}_1^T, \ldots, \mathbf{a}_{[d/2]}^T]$, $\mathbf{A}_2 = [\mathbf{a}_{[d/2]}^T, \ldots, \mathbf{a}_d^T]$. This decomposition allows for the approximation of high-rank matrices through a linear combination of lower-rank matrices. The same conclusion can be reached by beginning with a linear combination of rank-1 matrices. This forms the basis for a novel multi-head LoRA parameterization, which we will use as one baseline to compare with LoRA-the-Explorer.

> **Multi-head LoRA (MHLoRA)**  Given a matrix $\mathbf{W} \in \mathbb{R}^{m \times n}$, and constant $N$, multi-head LoRA parameterizes the weights as a linear combination of $N$ low-rank matrices $\mathbf{B}_n$ and $\mathbf{A}_n$:
>
> $$f_{\mathsf{mhlora}}(\mathbf{x}) = \mathbf{W}\mathbf{x} + \frac{s}{N} \sum_{n=1}^{N} \mathbf{B}_n \mathbf{A}_n \mathbf{x} \qquad (2)$$

Multi-head LoRA reparameterizes the full-rank weights into a linear combination of low-rank weights. Now we will point out that a single LoRA head can recover the performance of multi-head LoRA provided the single LoRA head is periodically merged into the full weights. Using the same rank $r$ for all the LoRA parameters, the dynamics of a single LoRA head (denoted with $\hat{\cdot}$) is equivalent to multi-head LoRA

$$\underset{\mathbf{B}_n \mathbf{A}_n}{\arg\min} \mathcal{L}\left(\mathbf{W} + \frac{s}{N} \sum_{n=1}^{N} \mathbf{B}_n \mathbf{A}_n\right) = \underset{\hat{\mathbf{B}}_n, \hat{\mathbf{A}}_n}{\arg\min} \mathcal{L}\left(\hat{\mathbf{W}} + \frac{s}{N} \hat{\mathbf{B}}_n \hat{\mathbf{A}}_n\right) \qquad (3)$$

when either $\sum_{n=1}^{N} \mathbf{B}_n \mathbf{A}_n$ is equal to $\hat{\mathbf{B}}_n \hat{\mathbf{A}}_n$, or when $\hat{\mathbf{W}} = \mathbf{W} + \frac{s}{N} \sum_{j \neq n}^{N} \mathbf{B}_j \mathbf{A}_j$; here we used a shorthand notation to indicate that sum is over all the LoRA parameters except for index $n$. Here we assume the parameters on both sides of the equation are initialized to be the same: $\mathbf{A}_n = \hat{\mathbf{A}}_n$ and $\mathbf{B}_n = \hat{\mathbf{B}}_n \forall n$. The first scenario is rank deficient, which we know is unable to recover the original model performance. The latter case necessitates that $\hat{\mathbf{W}}$ accumulates all the information of the LoRA parameters at every iteration. However, synchronization at every iteration can be expensive and a more practical choice is to use stale estimates of the LoRA parameters $\hat{\mathbf{W}} = \mathbf{W} + \frac{s}{N} \sum_{j \neq n}^{N} \mathbf{B}_j' \mathbf{A}_j'$ with $'$ indicating stale estimate of the parameters. This is equivalent to merging the LoRA parameters.

Merging every iteration ensure that the representation will not diverge from the intended update. While using stale estimates relaxes this equivalence, we observe that it can still match the standard training performance as shown in Table 1. Nevertheless, as the estimate becomes inaccurate, the optimization trajectory does indeed diverge from the optimization path of multi-head LoRA. We quantify this divergence in Figure 2. The divergence does not imply that the model won't optimize; rather, it suggests that the optimization trajectory will deviate from that of the multi-head LoRA.

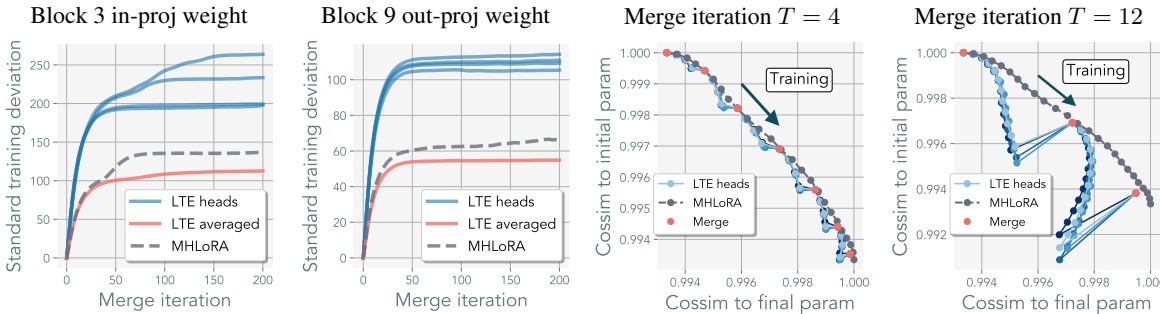

Figure 2: **Effects of merging LoRA heads. Left:** We measure $l_2$-norm deviation of the effective weights of multi-head LoRA (MHLoRA), and LoRA-the-explorer (LTE, our method) from the weights of standard training using ViT-S. We use 4 heads for both MHLoRA and LTE using the same initialization. We also plot the individual LoRA heads of LTE. These heads deviate more from standard training but their average closely follows that of MHLoRA. Depending on the merge iteration (x-axis), the estimation gap of using stale estimates is roughly the difference between the MHLoRA and LTE. The later the merge happens, the more LTE deviates from MHLoRA. **Right:** We project the dynamics of MHLoRA and LTE on to the parameters of MHLoRA. The y-axis is the initial parameters and x-axis is after training for 25 iterations. The projection is computed via computing the cosine similarity on the vectorized weights. We use merge iteration of 4 and 12 for LTE. The LTE projection closely follows that of trajectory arc of MHLoRA, where more frequent merges result in less deviation.

## 3.2 LoRA-the-Explorer

Our algorithm is designed with two primary considerations: (1) achieving an informative update $\Delta\mathbf{W}$ that does not require materialization of the full parameter size during training, and (2) parameterizing $\mathbf{W}$ such that it can be stored in low-precision and communicated efficiently. The latter can be achieved by using quantized weights and keeping a high-precision copy of $\mathbf{W}$.

We propose LoRA-the-Explorer (LTE), an optimization algorithm that approximates full-rank updates with parallel low-rank updates. The algorithm creates $N$-different LoRA parameters for each linear layer at initialization. Each worker is assigned the LoRA parameter and creates a local optimizer. Next, the data is independently sampled from the same distribution $\mathbf{x} = \{\mathbf{x}_1, \dots \mathbf{x}_N\}$. For each LoRA head $n$, the parameters are optimized with respect to its own data partition for $T$ iterations resulting in an update $\delta_{\mathrm{lora}_n} = -\eta \sum_{t=1}^{T} \nabla_{\mathrm{lora}_n} \mathbf{x}_i[t]$. We do not synchronize the optimizer state across workers. After the optimization, the resulting LoRA parameters are synchronized to compute the final update for the main weight $\Delta_{\mathrm{lora}}(\mathbf{x}) = \frac{1}{N} \sum_{n=1}^{N} \delta_n$. In the next training cycle, the LoRA parameters are re-trained with the updated weights $\mathbf{W}$. Since we do not train directly on the main parameter $\mathbf{W}$ we can use the quantized parameter $q(\mathbf{W})$ instead. Where one can either keep the high-precision weight only in the master node, or offload it from the device during training. This reduces not only the memory footprint of each worker but the also the transmission overhead. A pseudo-code is provided in Algorithm 1 and an illustration in Figure 3.

---

**Algorithm 1:** LoRA-the-Explorer (LTE)

**Input:** Dataset $\mathcal{D}_{\mathrm{train}}$, model $\mathcal{F}$, loss function $\mathcal{L}$
    parameters $\Theta = \{\mathbf{W}_0, \dots, \mathbf{W}_L\}$,
    merge scalar $s$, num workers $N$, merge iter $T$

---

**while** *not converged* **do**
    (*Optional*) quantize $\Theta$. Keep high-precision copy
    **(in parallel) for** *each worker* $n$ **do**
        **if** *LoRA not initialized* **then**
            $\mathcal{B}_n, \mathcal{A}_n \leftarrow$ **lora_parameterize**$(\mathcal{F})$;
        **end**
        **else**
            Reset parameters $\mathcal{B}_n$ to zero
            (*Optional*) reinitialize parameter $\mathcal{A}_i$
        **end**
        Optimize $\mathcal{B}_n, \mathcal{A}_n$ for $T$ iterations by minimizing
            $\mathbb{E}_{\mathbf{x},\mathbf{y} \sim \mathcal{D}_{\mathrm{train}}} [\mathcal{L}(\mathcal{F}(\mathbf{x}), \mathbf{y})]$
    **end**
    **for** *each worker* $n$ **do**
        # Synchronize by communicating LoRA parameters.
        **for** $\mathbf{B}_n, \mathbf{A}_n$ *in* $\mathcal{B}_n, \mathcal{A}_n$ **do**
            Merge LoRA params $\mathbf{W} \leftarrow \mathbf{W} + \frac{s}{n} \mathbf{B}_n \mathbf{A}_n$
        **end**
    **end**
**end**

---

Figure 3: **LTE diagram**: Our method is decomposed into 3 steps. (**1**) We parameterize the model with multiple LoRA heads and train them independently for $T$ iterations using different mini-batches sampled from the same distribution. This results in overall update of $\delta_{\mathrm{lora}_n}(\mathbf{x}) = -\eta \sum_t \nabla_{\mathrm{lora}_n}(\mathbf{x}[t])$ (**2**). Next, we accumulate the individual LoRA updates by averaging the heads $\Delta_{\mathrm{lora}}(\mathbf{x}) = \frac{1}{N}\sum_n \delta_{\mathrm{lora}_n}(\mathbf{x})$. (**3**) The update is applied to the main weights, and the LoRA parameter **B** is reset. The optimization repeats with the new LoRA parameters.

### 3.3 IMPLEMENTATION DETAILS

We discuss few of the implementation details we found necessary for improving the convergence speed and the performance of our method. Full training details and the supporting experiments can be found in Appendix A.

**Not resetting matrix A and optimizer states**    We investigate whether the matrices $\mathbf{A}_n$ would converge to the same sub-space during training. If so, it would necessitate resetting of matrices $\mathbf{A}_n$ or use of a regularizer. In Figure 5, we did not observe this to be the case. We observed the orthogonality of $\mathbf{A}$ to remain consistent through-out training and we found it to perform better without resets. We posit that re-learning matrix $\mathbf{A}$ and re-accumulating the optimizer state ends up wasting optimization steps. The comparison figures can be found in Appendix A.3 and more detailed discussion in Section 4.2.

**Scaling up $s$ and lowering learning rate $\eta$**

It is a common misconception that scaling $s$ has the same effect as tuning the learning rate $\eta$. During our experimentation we were unable to yield comparable performance when using standard value of $s$ (in the range of $1 \sim 4$). Instead, we found large value of $s$ and slightly lower learning rate $\eta$ to work the best. The standard practice is to set the scaling proportionately to the rank of the LoRA $s = \alpha/r$. This is done to automatically adjust for the rank (Hu et al., 2021). We use $\alpha = 4096$ ($s = 64$) and a learning rate of $\eta = 2 \cdot 10^{-4}$. It is worth noting that the learning-rate does not scale linearly with $s$ and the scalar only effects the forward computation (Appendix B.1). The scalar $s$ modifies the contribution of the LoRA parameters in the forward pass which has a non-trivial implication on the effective gradient. Moreover in Appendix B.2, we find that the update-rule moves in the direction that aligns $\mathbf{B}$ and $\mathbf{A}$, and scales quadratically with the learning rate.

**Significance of Initialization Strategies**

Initialization of LoRA plays a pivotal role in pre-training. Kaiming initialization used in the original work (Hu et al., 2021) – are not well-suited for rectangular matrices as discussed in (Bernstein et al., 2023; Yang & Hu, 2020). Given that LoRA parameterization often leads to wide matrices, alternative methods from (Bernstein et al., 2023) and (Glorot & Bengio, 2010) resulted in better empirical performance.

We use initialization scheme prescribed in (Bernstein et al., 2023) that utilizes a semi-orthogonal matrix scaled by $\sqrt{d_{out}/d_{in}}$. Note that these methods were originally designed for standard feed-forward models. Where as LoRA operate under the assumption that matrix $\mathbf{B}$ is zero initialized with residual connection. This aspect warrants further study for exact gain calculations. Our ablation studies, in Appendix A.3, indicate the best performance with Bernstein et al. (2023), with Kaiming and Xavier initializations performing similar. In ImageNet-1k, we found the performance gap to be more evident.

## 4    EXPERIMENTS

All training hyper-parameter details are held in Appendix A.1.

### 4.1    ITERATIVE LoRA MERGING

In Section 3.1, we motivated that iteratively merging LoRA parameters is a key component in accurately recovering the full-rank representation of the model. As a sanity check, in Appendix A.2 we assess the effectiveness of merging single LoRA head in the context of linear networks trained on synthetic least-squares regression datasets. The underlying rank of the optimal solution, $\mathbf{W}^*$, is controlled, and datasets are generated as $\mathbf{Y} = \mathbf{X}(\mathbf{W}^*)^\top$. Each $\mathbf{x} \in \mathbf{X}$ follows a normal distribution, $\mathbf{x} \sim \mathcal{N}(\mathbf{0}, \mathbf{I})$. Figure 8 evaluates the model's rank recovery across varying merge iteration $T$. Dimension of the weights $\mathbf{W}$ are set to $m = n = 32$. Without merging, the model performance plateaus rapidly on full-rank $\mathbf{W}^*$. In contrast, iterative merging recovers the ground truth solution with the rate increasing with higher merge frequency.

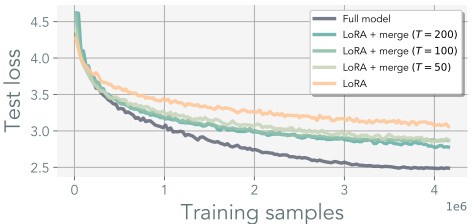

Figure 4: **Full-model vs. LoRA w/ merge:** ViT-S trained on ImageNet100. Merging and resetting the LoRA parameters achieves better performance than single-head LoRA pre-training but still cannot recover the standard full-model pre-training. Note that LoRA + merge is akin to concurrent work of ReLoRA (Lialin et al., 2023).

Further tests in Figure 4 using ViT-S (Dosovitskiy et al., 2020) with a patch-size of 32 on the ImageNet100 dataset (Tian et al., 2020) (a subset of ImageNet (Russakovsky et al., 2015)) confirm that merging of a single LoRA head outperforms standalone LoRA parameter training. However, frequent merging delays convergence, likely due to LoRA parameter re-initialization and momentum state inconsistencies. Additionally, the performance does not match that of fully trained models, indicating potential local minima when training with rank-deficient representations. We find that merge iteration of $T = 10$ to work the best for batch-size 4096, and $T = 20$ when using smaller batch-size of 256. With higher $T$ values, additional training may be required to achieve comparable performance (Stich, 2018; Wang & Joshi, 2021; Yu et al., 2019). Our initial efforts to improve merging using methods such as (Yadav et al., 2023) did yield better results. Nonetheless we believe with increased merge iteration, smarter merging techniques may be necessary. Existing literature in federated learning and linear-mode connectivity/model-averaging may provide insights in designing better merging criteria (refer to Section 5).

## 4.2 LoRA Parameter alignment

The efficacy of our optimization algorithm hinges on the ability of individual heads to explore distinct subspaces within the parameter space. We examine the extent to which data and initial parameters influence the intra-head similarity throughout training. In Figure 5, we compute the average cosine similarity is computed between the heads based on vectorized matrices $\mathbf{B}_n\mathbf{A}_n$. These tests were conducted with data samples drawn from the same distribution, and each set of LoRA parameters was exposed to a different set of samples. Dropout was disabled for these experiments.

Our results confirm that LoRA heads do not converge to the same representation. We find using different initialization across LoRA heads yields the greatest orthogonality. This orthogonality is further increased when different mini-batches are used. Importantly, the degree of alignment among LoRA heads remains stable post-initialization. In Appendix A.4, we find that lower cosine similarity to correspond well with model performance, where using different parameters and mini-batches to significantly outperform other configurations.

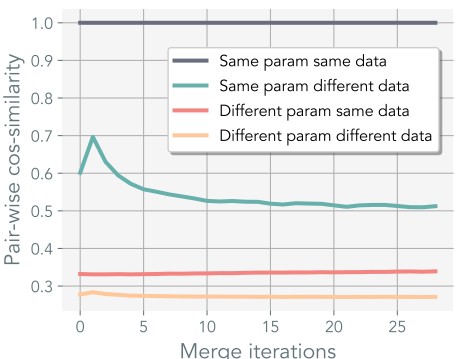

Figure 5: **LoRA alignment**: Alignment of LTE heads when varying parameters and data. Appendix A.4 provides the corresponding performance. "Different param" uses random initialization across each head, and "different data" uses different mini-batches. The similarity is computed on the first epoch of ImageNet100 on ViT-S. We use LTE of $r=8$ with 4 LoRA heads. Pair-wise similarity is averaged across all linear layers.

## 4.3 Impact of LoRA heads, Rank, and Merge Iteration on Performance

We systematically evaluate the effects of varying the number of LoRA heads, rank, and merge iteration on model performance for ImageNet100 in Table 1. Our findings indicate a monotonic improvement in performance with increased number of heads and rank. Conversely, extending the merge iteration negatively impacts performance. As in the case of least-squares regression, we found excessive merging to hurt model accuracy. With large

| | Model | Patch | LTE | Heads | Rank | Merge | Test loss ↓ | Test acc ↑ |
|---|---|---|---|---|---|---|---|---|
| | ViT-S | 32 | - | - | - | - | 1.78 | 71.97 |
| head | ViT-S | 32 | ✓ | 2 | 8 | 10 | 2.31 | 54.68 |
| | ViT-S | 32 | ✓ | 8 | 8 | 10 | 2.35 | 54.88 |
| | ViT-S | 32 | ✓ | 32 | 8 | 10 | 2.26 | 58.21 |
| | ViT-S | 32 | ✓ | 2 | 64 | 10 | 1.86 | 69.82 |
| | ViT-S | 32 | ✓ | 8 | 64 | 10 | 1.84 | 71.97 |
| | ViT-S | 32 | ✓ | 32 | 64 | 10 | 1.79 | 73.73 |
| rank | ViT-S | 32 | ✓ | 32 | 8 | 10 | 2.66 | 44.00 |
| | ViT-S | 32 | ✓ | 32 | 16 | 10 | 2.12 | 60.35 |
| | ViT-S | 32 | ✓ | 32 | 32 | 10 | 1.81 | 71.20 |
| | ViT-S | 32 | ✓ | 32 | 64 | 10 | 1.79 | 73.73 |
| | ViT-S | 32 | ✓ | 32 | 128 | 10 | 1.78 | 73.54 |
| merge | ViT-S | 32 | ✓ | 32 | 64 | 5 | 1.87 | 70.67 |
| | ViT-S | 32 | ✓ | 32 | 64 | 10 | 1.79 | 73.73 |
| | ViT-S | 32 | ✓ | 32 | 64 | 20 | 1.97 | 66.80 |
| | ViT-S | 32 | ✓ | 32 | 64 | 50 | 1.99 | 65.43 |
| | ViT-S | 32 | ✓ | 32 | 64 | 100 | 2.10 | 61.04 |

**Table 1: LTE ablation results on ImageNet100:** For *fixed cumulative training epoch* of 1200, we vary the number of heads, rank, and merge iteration of our method.

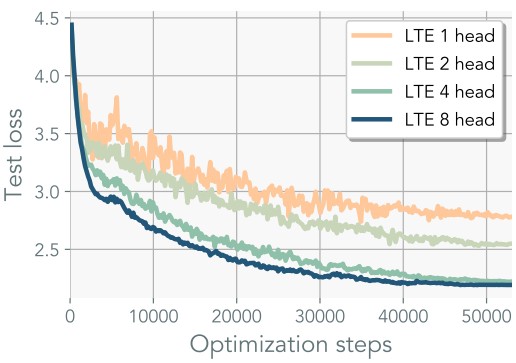

Figure 6: **LTE with same batch-size per head:** ViT-S trained on ImageNet100. We use the same batch-size for each LoRA head with rank $r = 8$. In contrast to other figures, we plot the loss in optimization steps. LTE with more heads converge to better solution but require longer training samples to converge.

enough rank and head, we found the model to converge to better test accuracy, even if the test loss was similar. We hypothesize it averaging of the LoRA heads have a regularization effect similar to that model ensembling.

We use ViT-S as the primary architecture for analysis, which has a hidden dimension of 384 and an MLP dimension of 1536. We find that setting the product of the number of heads and the rank of the LoRA larger than the largest dimension of the model serves as a good proxy for configuring LTE. For example, using 32 heads with $r = 64$ results in $2048 > 1536$. However, when it comes to increasing the number of heads rather than rank, we noticed longer training iterations were required to achieve comparable performance. We discuss a potential cause in the slowdown in convergence in the preceding section.

### 4.4 GRADIENT NOISE WITH PARALLEL UPDATES

In our ablation study, we utilized a fixed cumulative batch size of 4096 and a training epoch of 1200. Each LoRA head received a reduced batch size of $\frac{4096}{\text{heads}}$. Our findings indicate that scaling the rank exerts a greater impact than increasing the number of heads. Due to the proportional scaling of gradient noise with smaller mini-batches (Smith & Le, 2017), we hypothesize that gradient noise is the primary factor contributing to slower convergence, in addition to the use of stale parameter estimates. To validate this hypothesis, we employed the same mini-batch size across all heads in Figure 6, using a reduced rank of $r = 8$. When we adjusted the batch size in proportion to the number of heads and measured it with respect to the optimization steps, the impact of varying the number of heads became more pronounced. While increasing the number of heads necessitates more sequential FLOPs, it offers efficient parallelization. Furthermore, using a larger batch size for gradient estimation may prove beneficial in distributed training, as it increases the computational workload on local devices. Careful optimization of the effective batch size to maximize the signal-to-noise ratio may be crucial for achieving maximum FLOP efficiency.

Furthermore, the results presented in Figure 10 align with our gradient noise hypothesis. Models trained with identical mini-batches demonstrate improved performance as the learning rate was lowered by the scheduler. Using the same mini-batch across multiple LoRA heads may help mitigate the gradient noise near convergence.

### 4.5 PERFORMANCE SCALING ON IMAGENET-1K

We scaled up our method to ImageNet-1K. We followed the training protocols detailed in Appendix A.1. In accordance with our initial hypothesis on gradient noise, we doubled the batch size to 8192 (see Appendix A.7). Since using different mini-batches was crucial early in training, we did not alter the way mini-batches were sampled. Scheduling the randomness for the mini-batches is an option we have not yet explored.

In the initial training phase, we observed that LTE outperformed standard training. However, as training approached completion, standard training overtook LTE, necessitating additional iterations for LTE to achieve

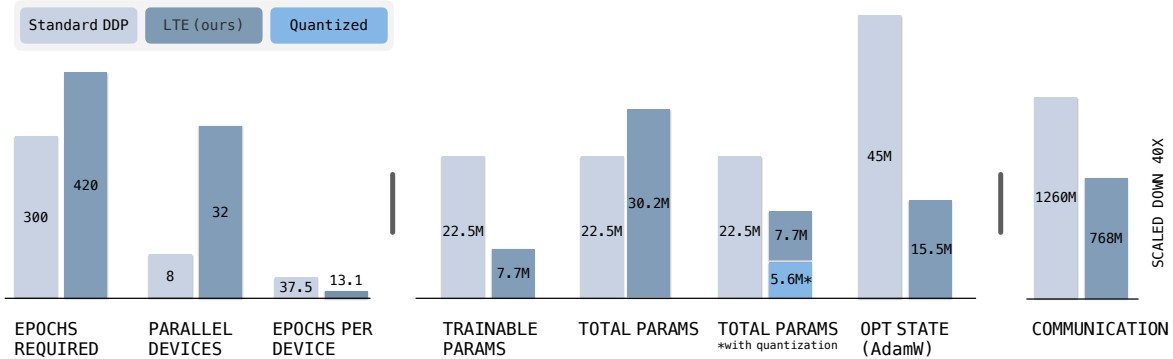

Figure 7: **ImageNet compute analysis**: We break down the hypothetical computational cost for training ViT-S on ImageNet1k. We compare against distributed data-parallel with 8 devices. LTE requires 40% longer to achieve the same performance of 68% top-1 accuracy on 1000-way classification. We used 32 LoRA head with $r = 64$. Our method requires fewer trainable parameters per device. This in turn could enable fitting of larger models in low-memory hardware, with increased parallelization. With smaller memory footprint and infrequent communication our method uses lower total communication cost. Further discussion is in Section 4.5.

comparable performance. Standard training appeared to benefit more from a smaller learning rate compared to LTE. For ViT-S, the model took 40% longer to converge to the same top-1 accuracy of 68% (see Appendix A.6).

The primary focus of our work was to investigate whether it is possible to train deep networks with parallel low-rank adapters; hence, we did not aim to maximize efficiency. However, we do provide a hypothetical computation analysis for future scaling efforts. Let the model size be denoted by $M_{\text{ddp}} = M$, and $M_{\text{lte}}$ for LTE, and the respective number of devices for each method be denoted with $N_{\text{ddp}}$, and $N_{\text{lte}}$ . With quantization, each LTE device would require a memory footprint of $qM + M_{\text{lte}}$. With base model operating in 16-bit precision, using 4-bit quantization results in $q = 0.25$. With AdamW, DDP necessitates an additional $2M$ parameters, making the total memory footprint $3M$ per device. For LTE, the total memory footprint per device is $qM + 3M_{\text{lte}}$. Assuming the training is parameter-bound by the main weights $r \ll \min(m, n)$, LTE can leverage GPUs that are roughly $1/3$ the size required for DDP. It is worth noting that LTE requires 40% more data to train, and a slowdown of 20% per iteration when using quantization methods such as QLoRA. If the cost of low-memory devices is lower, these slowdowns may be negligible compared to the speed-up achieved through parallelization. On average, each LTE device observes $1/3$ less data than a device in DDP. With improvements in our method and future advances in quantization, we believe this gap will reduce. The compute analysis for ViT-S on ImageNet1k is illustrated in Figure 7.

Communication also presents bottlenecks when training models across nodes. For a single node, one can interleave the communication of gradients asynchronously with the backward pass (Li et al., 2020). In multi-node systems, the communication scales with the size of the trained parameters, often bottlenecked by interconnect speed, especially when high-throughput communication hardware, such as InfiniBands, are not utilized. Using standard all-reduce, gradient is shared between each device for a total communication of $N_{\text{ddp}}(N_{\text{ddp}} - 1)M$. For LTE we communicate every $T$ iteration hence we have $\frac{1}{T}N_{\text{lte}}(N_{\text{lte}} - 1)M$. To maximize efficacy of LTE, an alternative approach is to use a parameter server for 1-and-broadcast communication. Here gradients are sent to the main parameter server and averaged. The accumulated updates are broadcasted back to other nodes. DDP with a parameter server would use $2(N_{\text{ddp}} - 1)M$ and LTE would use $\frac{1}{T}((N_{\text{lte}} - 1)M_{\text{lte}} + (N_{\text{lte}} - 1)qM)$. Moreover, LTE can leverage lower-bandwidth communication since the parameters shared between devices are strictly smaller by a factor of $M_{\text{ddp}}/M_{\text{lte}}$.

## 5 RELATED WORKS

**Training with adapters**    The use of LoRA has garnered considerable attention in recent research. While our work focuses on using adapters to pre-train model from scratch, majority of the works have focused on enhancing fine-tuning processes through these adapters (Chavan et al., 2023; Zhang et al., 2023b), while others aim to diminish computational requirements (Zhang et al., 2023a) or to offset part of the pre-training computation (Lialin et al., 2023). Moreover Wang et al. (2022) explores the use mixture-of-adapters (MoE-variant) for parameter efficient fine-tuning and uses "model-soup"-style (Wortsman et al., 2022b) averaging for efficient inference. Various forms of adapters have been proposed in prior research, each serving specific applications.

Additive adapters (Zhang et al., 2021) augment the model size for inference, and hence the community has turned towards linear adapters either in the form residual connections (Cai et al., 2020) or affine parameters in batch-norm, (Bettelli et al., 2006; Mudrakarta et al., 2018). Adapters have been applied for natural language processing (Houlsby et al., 2019; Stickland & Murray, 2019), video (Yang et al., 2023; Xing et al., 2023), computer vision (Sax et al., 2020; Zhang & Agrawala, 2023; Chen et al., 2022b), incremental learning (Rosenfeld & Tsotsos, 2018), domain adaptation (Rebuffi et al., 2018), and vision-language tasks (Gao et al., 2023; Radford et al., 2021; Sung et al., 2022), text-to-vision generative models (Mou et al., 2023), and even perceptual learning (Fu et al., 2023).

**Distributed Training and Federated Learning**  Our work has relevance to both distributed and federated learning paradigms, wherein each head is conceptualized as a distinct computational device. Federated learning addresses various topics including low-compute devices, high-latency training, privacy, and both cross and in-silo learning, as comprehensively discussed in (McMahan et al., 2017; Wang et al., 2021). Communication efficiency serves as a cornerstone in both distributed and federated learning. Techniques such as *local steps* have been employed to mitigate communication load (Lin et al., 2018; Povey et al., 2014; Smith et al., 2018; Su & Chen, 2015; Zhang et al., 2016). These methods defer the averaging of weights to specific optimization steps, thus alleviating the communication cost per iteration. The effectiveness of decentralized training have been studied in (Lian et al., 2017; Koloskova et al., 2019; 2020; Coquelin et al., 2022). Traditionally, activation computations have dominated the computational load. However, the advent of gradient checkpointing (Chen et al., 2016) and reversible gradient computation (Gomez et al., 2017; Mangalam et al., 2022) has shifted the training process toward being increasingly parameter-bound. Techniques such as gradient or weight compression also seek to reduce the communication burden (Lin et al., 2017; Aji & Heafield, 2017; Wen et al., 2017). Combining models in federated learning is often credited to FedAvg (McMahan et al., 2017). Numerous studies explore the use of weighted averaging to improve convergence speed (Li et al., 2019). Since then, many works have tried to use probabilistic frameworks to understand and improve merging (Hsu et al., 2019; Wang et al., 2019; Reddi et al., 2020). The conditions for optimal merging is still an open question, with recent efforts to improve updating with stale parameters has been explored in Chen et al. (2022a). Server momentum and adaptive methods constitute another active area of research. Where macro synchronization steps may be interpreted as gradients, thus facilitating bi-level optimization schemes (Hsu et al., 2019; Wang et al., 2019; Reddi et al., 2020). Initial efforts to employ federated learning with large models have been made. Yuan et al. (2022) examined the cost models for pre-training Language Learning Models (LLMs) in a decentralized configuration. Wang et al. (2023) suggested the utilization of compressed sparse optimization methods for efficient communication.

**Linear mode connectivity and model averaging**  Linear mode connectivity (Garipov et al., 2018) pertains to the study of model connectivity. Deep models are generally linearly disconnected but can be connected through nonlinear means (Freeman & Bruna, 2016; Draxler et al., 2018; Fort & Jastrzebski, 2019). Under the same initialization, linear paths with constant energy exists in trained models (Nagarajan & Kolter, 2019; Frankle et al., 2020; Wortsman et al., 2022b). For models with different initializations, parameter permutations can be solved to align them linearly (Brea et al., 2019; Tatro et al., 2020; Entezari et al., 2021; Simsek et al., 2021). Following this line research, numerous works have delve into model averaging and stitching. Where averaging of large models have shown to improve performance (Wortsman et al., 2022b; Ainsworth et al., 2022; Stoica et al., 2023; Jordan et al., 2022). Model stitching (Lenc & Vedaldi, 2015) has also shown to yield surprising transfer capabilities (Moschella et al., 2022). This idea is conceptually related to optimal averaging in convex problems (Scaman et al., 2019) and the "Anna Karenina" principle where successful models converge to similar solutions (Bansal et al., 2021). The effectiveness of averaging models within ensembles is well-established (Huang et al., 2017; Izmailov et al., 2018; Polyak & Juditsky, 1992). Utilizing an average model as a target has also been investigated (Tarvainen & Valpola, 2017; Cai et al., 2021; Grill et al., 2020; Jolicoeur-Martineau et al., 2023; Wortsman et al., 2022a).

# 6  CONCLUSION

In this work, we investigated the feasibility of using low-rank adapters for model pre-training. We introduced LTE, a bi-level optimization method that capitalizes on the memory-efficient properties of LoRA. Although we succeeded in matching performance on moderately sized tasks, several questions remain unresolved. These include: how to accelerate convergence during the final $10\%$ of training; how to dynamically determine the number of ranks or heads required; whether heterogeneous parameterization of LoRA is feasible, where each LoRA head employs a variable rank $r$; and leveraging merging strategies to accompany higher local optimiza-

tion steps. Our work serves as a proof-of-concept, demonstrating the viability of utilizing low-rank adapters for neural network training from scratch. However, stress tests on larger models are essential for a comprehensive understanding of the method's scalability. Addressing these open questions will be crucial for understanding the limitations of our approach. We anticipate that our work will pave the way for pre-training models in computationally constrained or low-bandwidth environments, where less capable and low-memory devices can collaboratively train a large model, embodying the concept of the "*wisdom of the crowd.*"

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

# A APPENDIX

## A.1 TRAINING DETAILS

**Training details:** We adhere to the standard training protocols. Below are the references:

| | |
|---|---|
| Vision training code | PyTorch TorchVision references |
| ViT implementation | PyTorch TorchVision models |
| MLP-mixer implementations | huggingface/pytorch-image-models |
| Quantization | TimDettmers/bitsandbytes |

We replace the fused linear layers with standard linear layers to use LoRA. LoRA is applied across all linear layers. All experiments incorporate mixed-precision training. For nodes equipped with 4 GPU devices, we implement gradient checkpointing. Gradient checkpointing is also utilized for ViT-L models.

**Hardware:** Our experiments were conducted using various NVIDIA GPUs, including V100 and Titan RTX.

**Architecture detail:**

| Architecture | ViT-S | ViT-B | ViT-L |
|---|---|---|---|
| Patch-sizse | 32 | 32 | 32 |
| Attention blocks | 12 | 12 | 24 |
| Attention heads | 6 | 12 | 16 |
| Hidden dim | 6 | 768 | 1024 |
| MLP dim | 1536 | 3072 | 4096 |
| Total parameters | 22.9M | 88.2M | 306.5M |

Table 2: ViT architecture details.

| Architecture | Mixer-T | Mixer-S | Mixer-B |
|---|---|---|---|
| Patch-sizs | 32 | 32 | 32 |
| Mixer blocks | 6 | 8 | 12 |
| Embed dim | 384 | 512 | 768 |
| MLP dim | 1536 | 2048 | 3072 |
| Total parameters | 8.8M | 19.1M | 60.3M |

Table 3: MLP-mixer architecture details

| Architecture | NanoGPT | GPT2 |
|---|---|---|
| Block-size | 256 | 1024 |
| Attention blocks | 6 | 12 |
| Attention heads | 6 | 12 |
| Hidden dim | 384 | 768 |
| MLP dim | 1152 | 2304 |
| Total parameters | 10.7M | 124.4M |

Table 4: LLM GPT architecture details.

**Dataset details:**

| Dataset | CIFAR10 | CIFAR100 | STL10 | CALTECH256 | SUN397 | ImageNet100 | ImageNet1K |
|---|---|---|---|---|---|---|---|
| Original image-size | 32x32 | 32x32 | 96x96 | Variable | Variable | Variable | Variable |
| Training image-size | 224x224 | 224x224 | 224x224 | 224x224 | 224x224 | 224x224 | 224x224 |
| Number of classes | 10 | 100 | 10 | 257 | 397 | 100 | 1,000 |
| Number of images | 60,000 | 60,000 | 13,000 | 30,607 | 108,754 | 130K | >1.2M |
| Learning-rate $\eta_{\text{default}}$ | $3 \cdot 10^{-4}$ | $3 \cdot 10^{-4}$ | $3 \cdot 10^{-4}$ | $3 \cdot 10^{-4}$ | $3 \cdot 10^{-3}$ | $3 \cdot 10^{-3}$ | $3 \cdot 10^{-3}$ |
| Learning-rate $\eta_{\text{lte}}$ | $2 \cdot 10^{-5}$ | $2 \cdot 10^{-5}$ | $2 \cdot 10^{-5}$ | $2 \cdot 10^{-5}$ | $5 \cdot 10^{-6}$ | $3 \cdot 10^{-5}$ | $3 \cdot 10^{-5}$ |
| Batch-size | 1024 | 1024 | 1024 | 1024 | 1024 | 4096 | 8192 |

Table 5: Specifications for vision datasets. Most images in variable size datasets are larger than the training image-size. We provide training configuration used for ViT. For MLP-Mixer on ImageNet100, we use learning rate of 0.001 for full-model pre-training and $1 \cdot 10^{-4}$ for LTE, both with a batch-size of 4096.

| Dataset | Shakespeare | TinyStories |
|---|---|---|
| Total number of tokens | 1.0M | 474.0M |
| Tokenizer size | 65 | 50304 |
| Learning-rate $\eta_{\text{default}}$ | $1 \cdot 10^{-3}$ | $6 \cdot 10^{-4}$ |
| Learning-rate $\eta_{\text{lte}}$ | $2 \cdot 10^{-6}$ | $5 \cdot 10^{-5}$ |
| Batch-size | 512 | 512 |
| Block-size | 256 | 1024 |

Table 6: Specifications for LLM datasets and hyper-parameters used for miniGPT on Shakespeare and GPT2 on Tinystories.

**LTE Optimization Details:** We use $\alpha = 4096 \sim 8192$, which is $s = 128 \sim 256$ when $r = 32$ and $s = 64 \sim 128$ when $r = 64$. A good rule of thumb for the learning rate is to set it at approximately $0.1 \sim 0.05\times$ the standard model training learning rate. We use the same learning rate scheduler as the standard pre-training, which is cosine learning-rate decay with linear warmup.

**LTE Batching Detail:** We use a fixed cumulative batch size for LTE. This means that given a batch size $B$ with $N$ LoRA heads, each head receives a batch size of $[B/N]$. When counting the training iterations, we count $B$ and not $[B/N]$. Counting using $[B/N]$ would significantly inflate our number, overselling our method. LTE training epochs were set to $4\times$ the cumulative batch size, and we exit early when we match the performance of full-model training. For smaller datasets, our method seemed to consistently outperform the baseline, likely due to the regularization properties of rank and over-parameterization.

**LTE Implementation Details:** We implemented LTE using Parameter Server and PyTorch DDP. While the former is theoretically more beneficial for our method, we conducted most of our development using the latter due to its well-optimized backend that does not require rewriting communication logic. We utilize `torch.vmap` and simulate multiple devices on the same GPU.

**LTE for Convolution, Affine, and Embedding Layers:** Specific choices for all these layers did not seem to make a significant impact on the final performance, but we detail the choices we made below.

For convolution layers, we use the over-parameterization trick in (Huh et al., 2023), which uses $1 \times 1$ for the second layer. Since convolution layers are typically used at most once in the models we tested, we did not explore beyond this parameterization. However, there are other potential choices for low-rank parameterization of convolution layers, such as channel-wise convolution and separable convolutions.

For affine parameters, there is no notion of low-rank decomposition, but it is used in normalization layers. We tried various strategies to train and communicate these parameters, all resulting in comparable performance. For affine parameters, we tried: (1) LoRA-style vector-vector parameterization $\mathbf{a}, \mathbf{b}$, (2) LoRA-style vector-scalar parameterization $\mathbf{A} = \mathbf{a}, b$, (3) DDP-style averaging, and (4) removing affine parameters. We use vector-scale parameterization for the experiments in the main paper.

Lastly, for the embedding layer, we found that using the standard averaging technique or allowing only one model to train the embedding layer worked best. We chose to use standard averaging at the same iteration as the rest of the LoRA layers.

### A.2    MERGE WITH LEAST-SQUARES

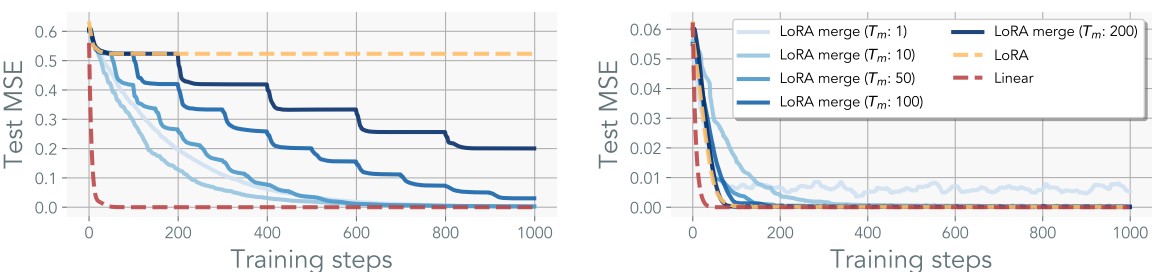

Figure 8: **Least-squares with LoRA**: Linear models parameterized with LoRA with varying target rank. Least-squares with $\mathbb{R}^{32\times32}$, with target rank of 32 (**right**) and 8 (**left**). LoRA is parameterized with rank $r = 4$. With merging, the model can recover the solution, with convergence scaling with merge frequencies.

We train a linear networks, parameterized with LoRA, on least-squares regression. Here we artificially constructed the problem to control for the underlying rank of the solution $\mathbf{W}^*$. We then constructed a dataset by randomly generating $\mathbf{Y} = \mathbf{X}(\mathbf{W}^*)^{\mathsf{T}}$. Where for each element $\mathbf{x} \in \mathbf{X}$ is drawn from a normal distribution $\mathbf{x} \sim \mathcal{N}(\mathbf{0}, \mathbf{I})$.

Figure 8 visualizes the model's ability to recover the underlying ground-truth solution across various merge iterations $T$. Here, the optimal solution $\mathbf{W}^*$ is set to be full rank where $m = n = 32$. We employ a naive re-initialization strategy of initializing $\mathbf{A}$ with a uniform distribution scaled by the fan-out.

Without merging the LoRA parameters, the model's performance rapidly plateaus. In contrast, models trained with merges can eventually recover the full-rank solution, with the recovery rate scaling with the frequency of merges.

### A.3 ABLATION

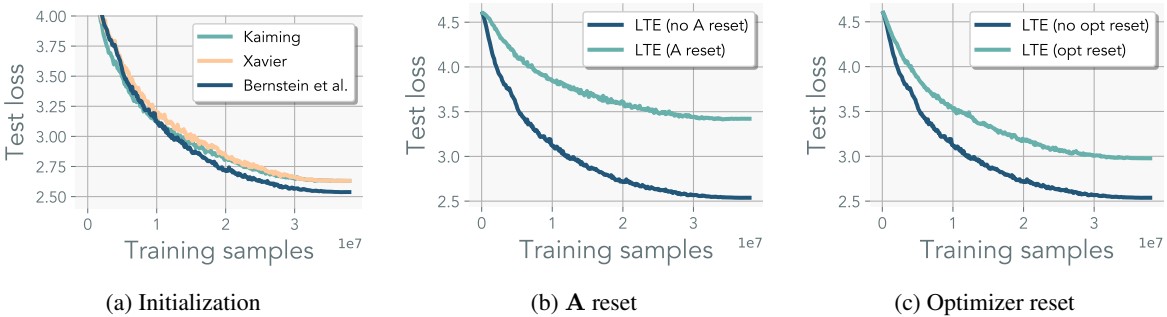

|  (a) Initialization | (b) **A** reset | (c) Optimizer reset |

Figure 9: **Ablation**: These models were trained using ViT-S with 8 heads with rank $r = 16$. (**Left**) different initialization scheme. (**Middle**) resetting **A**. (**Right**) resetting optimizer states for both **A** and **B**.

We conducted an ablation study focusing on initialization, resetting of LoRA **A**, and resetting the optimizer for the LoRA parameters. All ablation studies presented here were conducted with a LTE with rank $r = 16$, 8 heads using ViT-S on ImageNet100.

Kaiming initialization serves as the default scheme for LoRA. The conventional Kaiming initialization is tailored for square matrices and is dependent solely on the input dimension. Given that LoRA parameters often manifest as wide matrices, we experimented with various initialization schemes. Xavier initialization (Glorot & Bengio (2010)) preserves the variance relationship of a linear layer for rectangular matrices. Bernstein et al. (Bernstein et al. (2023)) employ semi-orthogonal initialization to preserve the spectral norm of the input. As depicted in Figure 9a, the method by Bernstein et al. proved most effective. It should be noted that all these initialization methods do not assume residual connection or zero-ed out LoRA parameters. Hence, further tuning of the gain parameters might be needed.

When resetting the LoRA parameters, we investigated the impact of resetting matrix **A** as well as its optimizer. As shown in Figure 9b, we found that resetting matrix **A** adversely affects model performance, possibly due to the necessity of relearning the representation at each iteration and discarding the momentum states. In Figure 9c, we also experimented with retaining the LoRA parameters while resetting the optimizer state for those parameters. Similarly, we found that resetting the optimizer state diminishes performance.

## A.4 THE EFFECT OF RANDOMNESS IN LORA PARAMETERS AND DATA

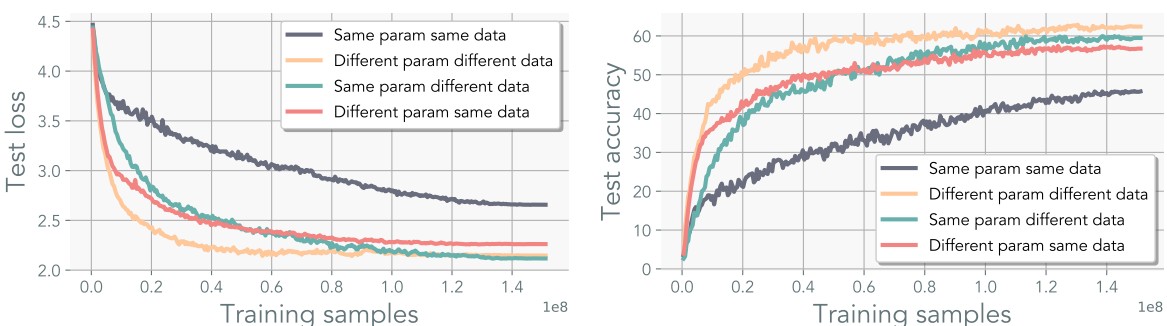

Figure 10: **The effect of training with different mini-batches and LoRA parameters**

In Section 4.2, we observed that variations in mini-batches and LoRA head initialization influenced the cosine similarity between the heads. In this section, we trained the model to convergence using these configurations. The model either utilized the same or different mini-batches drawn from an identical distribution. We also initialized the LoRA parameters to be either the same or different. Our findings indicate that cosine similarity serves as a reliable metric for evaluating model performance. However, models employing identical data benefited more when the learning rate was reduced. This suggests that minimizing gradient noise by gradually using similar mini-batches is crucial. Alternatively, one could incrementally increase the batch size.

## A.5  GRASSMAN AND COSINE DISTANCE OF LoRA HEADS

We measure cosine and Grassman distance of the LoRA heads to measure orthogonality between the optimized sub-spaces. Grassman distance measures the distance of two linear subspaces and is defined as:

> **Grassman distance**  Given subspaces $U$ and $V$, and given the singular values $U^T V = X \Sigma Y^T$, where $\Sigma$ is a diagonal matrix with singular values $\sigma_i$. The principal angles $\theta_i$ between $U$ and $V$ are given by $\theta_i = \cos^{-1}(\sigma_i)$. Then the Grassmann distance $d_{\mathsf{Grassman}}(U, V)$ is then defined as:
>
> $$d_{\mathsf{Grassman}}(U, V) = \left( \sum_{i=1}^{k} \theta_i^2 \right)^{\frac{1}{2}}$$
>
> where $k$ is the number of principal angles, typically the dimension of the smaller subspace.

For LoRA the number of principal components will most likely be spanned by the LoRA rank $r$. The pairwise Grassman distance is measured by:

$$\frac{1}{2N} \sum_{(i,j) \in [1,N] \wedge i \neq j} d_{\mathsf{Grassman}}(\mathbf{B}_i \mathbf{A}_i, \mathbf{B}_j \mathbf{A}_j) \tag{4}$$

The Grassmann distance measures how different two subspaces is by summing over the principal angles. This metric was used in (Hu et al., 2021). For cosine distance, the angle is measured on the vectorized parameters and measures the angle/alignment between the updates.

Figure 11 plots the is computed on ViT-S with 4 LoRA heads with rank $r = 32$. For both metrics, we measure the average pairwise distance of the LoRA heads using LTE with $T = 10$. We plot the distance for 6 linear layers throughout model, and also plot the average of all layers. We observe consistent orthogonality between each heads throughout optimization for both measures.

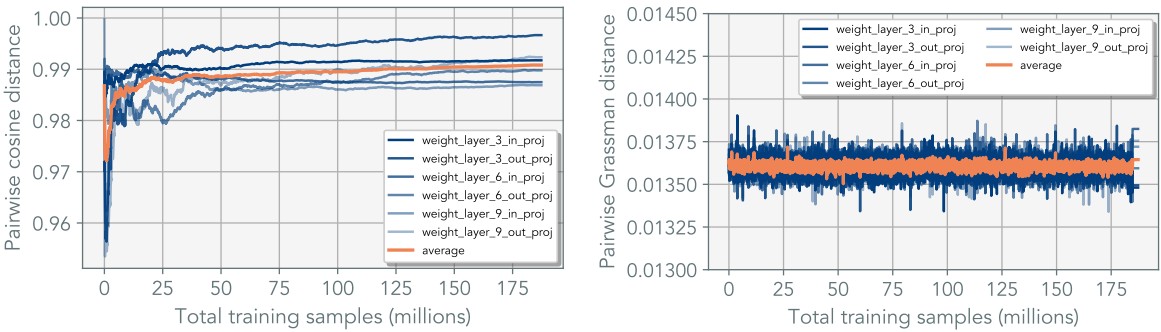

Figure 11: **Cosine and Grassman distance of LTE.**

## A.6 TRAINING CURVE ON IMAGENET1K

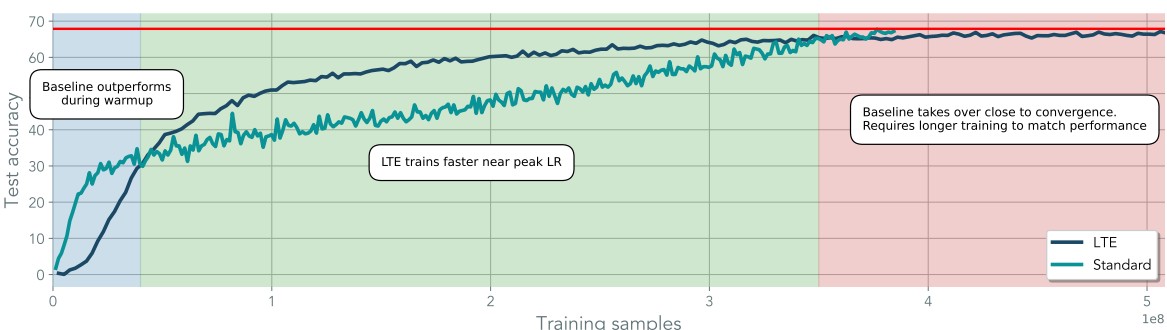

Figure 12: **ImageNet1k test curve**

We plot the test curves in Figure 12 for both standard and LTE training on ImageNet-1K, using a cosine learning rate scheduler for both. Training LTE for 300 epochs performed roughly 5% worse. Hence, we repeated the experiment by setting training epoch to 600. The final performance was matched at around 420 epochs. LTE was also trained with doubled batch size of 8192. Early in the training phase, the baseline outperformed LTE, but this trend was quickly reversed after first initial epochs. LTE approached its final performance quite rapidly. However, LTE fell short when compared to the standard training duration by 300 epochs, and additional 120 epochs were required to reach the same test accuracy. Unlike LTE, we found that standard training benefited significantly from small learning rate. Although we posit that gradient noise and stale parameter estimates are the primary cause of this gap, further investigation is required. We observed this trend across all ViT sizes. Few potential way to mitigate the slow convergence may be to synchronize the mini-batches or the LoRA parameters as the model is trained.

## A.7 EFFECT OF BATCH-SIZE ON IMAGENET1K

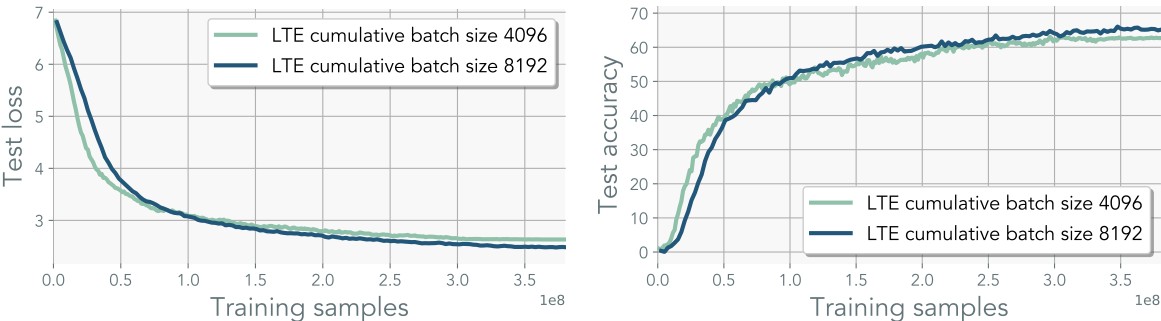

Figure 13: **ImageNet1k with doubled accumulative batch-size**

Utilizing larger batch sizes is beneficial for maximizing FLOP efficiency. However, when evaluated in terms of samples seen, larger batch sizes are known to underperform Masters & Luschi (2018). Given that LTE introduces more noise, we hypothesized that a reduced learning rate could be adversely affected by both gradient noise and estimate noise from using merges. In Figure 13, we experimented with increasing the batch size and observed a moderate improvement of 3% with bigger batches.

# B   RANK OF THE MODEL IN PRE-TRAINING

Figure 14: **Rank dynamics of ViT for standard training and LTE**. Rank is measured using effective rank. We track the rank of the weights and update to the main weight throughout training.

We measure the effective rank (Roy & Vetterli, 2007) of standard training and LTE throughout training.

> **(Definition) Effective rank (spectral rank)**  For any matrix $A \in \mathbb{R}^{m \times n}$, the effective rank $\rho$ is defined as the Shannon entropy of the normalized singular values:
>
> $$\rho(A) = \exp\left(-\sum_{i=1}^{\min(n,m)} \bar{\sigma}_i \log(\bar{\sigma}_i)\right),$$
>
> where $\bar{\sigma}_i = \sigma_i / \sum_j \sigma_j$ are normalized singular values, such that $\sum_i \bar{\sigma}_i = 1$.

The rank of the updates for standard training are the gradients $\nabla_{\mathbf{W}}\mathcal{L}$, and for LTE its $\frac{s}{N}\sum_n \mathbf{B}_n \mathbf{A}_n$. In the context of standard training, the rank of the weights exhibits only a minor decrease throughout the optimization process. Conversely, the rank of the gradient monotonically increases following the initial epochs. This observation serves as an empirical evidence that approximating the updates with a single LoRA head is not feasible. LTE, despite its markedly different dynamics, has the capability to represent full-rank updates throughout the training period. This may also be useful for designing how many LoRA heads to use with LTE, where the number of LoRA heads can start with one and slowly annealed to match the maximum rank of the weights.

## B.1 IS SCALING $s$ THE SAME AS SCALING THE LEARNING RATE?

There is a misconception that the scalar $s$ only acts as a way to tune learning-rate in these updates. Focusing on the update for $\mathbf{B}$ (same analysis holds for $\mathbf{A}$), we can write out the gradient as:

$$g(\mathbf{B}) = \frac{\partial \mathcal{L}}{\partial \mathbf{B}} = s \frac{\partial \mathcal{L}}{\partial \mathbf{z}_{out}} (\mathbf{A}\mathbf{z}_{in})^T = s\bar{g}_t \tag{5}$$

If we were using stochastic gradient descent, we would expect $s$ to behave like a linear scaling on the learning rate:

$$\Delta(\mathbf{B}) = -\eta s \bar{g} \tag{6}$$

Where we denoted $\bar{g}$ as the component of the gradient with $s$ factored out. We now show that $s$ does not linearly scale the learning rate for Adam (this analysis can be extended to scale-invariant optimizers). Using the same notation used in , Adam is a function of the first-order momentum $m_t$ and second-order momentum $v_t$. One can factor out $s$ from the momentum term: $m_t = s\hat{m}_t = s\beta_1\hat{m}_{t-1} + (1 - \beta_1)\hat{g}_t$ and $v_t = s^2\hat{v}_t = s^2\beta_1\hat{v}_{t-1} + (1 - \beta_1)\hat{g}_t^2$. Incorporating the gradient in to the update rule we see that the adaptive update does not depend linearly on $s$:

$$\Delta(\mathbf{B}) = -\eta \frac{m_t}{\sqrt{v_t} + \epsilon} = -\eta \frac{s\hat{m}_t}{\sqrt{s^2\hat{v}_t} + \epsilon} = -\eta \frac{\hat{m}_t}{\sqrt{\hat{v}_t} + \epsilon} \tag{7}$$

However, $\hat{g}_t$ is not invariant to $s$. Therefore, while $s$ is not the same as the learning rate, it will impact the downward gradient $\frac{\partial \mathcal{L}(\dots,s)}{\partial \mathbf{z}_{out}}$. We discuss this in the next sub-section. It is worth noting that $s$ quadratically impacts the attention maps. Consider a batched input $\mathbf{X}$ with the output of a linear as $\hat{\mathbf{X}} = \mathbf{W}\mathbf{X} + s\mathbf{B}\mathbf{A}\mathbf{X} = \mathbf{W}\mathbf{X} + s\mathbf{D}$. Then un-normalized attention map is dominated by the LoRA parameters:

$$\left(\mathbf{W}_Q\hat{X}\right)\left(\mathbf{W}_K\hat{X}\right)^T = \mathbf{W}_Q\left(\mathbf{W}\mathbf{X} + s\mathbf{D}\right)\left(\mathbf{X}^T\mathbf{W}^T + s\mathbf{D}^T\right)\mathbf{W}_K^T \tag{8}$$

$$= \cdots + s^2\mathbf{W}_Q\mathbf{D}\mathbf{D}^T\mathbf{W}_K^T \tag{9}$$

Unlike learning rate, scaling $s$ affects both the forward and backward dynamics. Large $s$ emphasizes the contribution of the LoRA parameters, which may explain why we have observed better performance when using larger $s$ for pre-training. It is possible that using a scheduler for $s$ could further speed up training, or even better better understand how to fuse $s$ into the optimizer or $\mathbf{A}$; we leave this for future work. Next, we dive into the effect of $s$ on $\bar{g}_t$.

## B.2 THE EFFECTIVE UPDATE RULE FOR LoRA IS DIFFERENT FROM STANDARD UPDATE

**Effective update of LoRA.** Let $\mathbf{W}$ be the original weight of the model, and denote $g(\mathbf{W}) = g$ as the gradient of the parameter. Let $\hat{\mathbf{W}} = \mathbf{W} + s\mathbf{BA}$ be the effective weight of the LoRA parameterization, and $g(\hat{\mathbf{W}}) = \hat{g}$ be its corresponding effective gradient. Then the LoRA parameterization is related to the gradient of the standard parameterization by

$$\hat{g} = s\left(\mathbf{BB}^T g - g\mathbf{A}^T\mathbf{A}\right) - s^2\eta\left(g\left(\mathbf{BA}\right)^T g\right) \tag{10}$$

When $s$ is small, we can safely discard the second term as it will scale quadratically with learning rate $\eta\hat{g}$. However, when $s$ is large, the contribution of the second term becomes non-negligible. This term can be interpreted as the alignment of the LoRA parameters, and taking a step in this direction encourages $\mathbf{B}$ and $\mathbf{A}$ to be spectrally aligned. The increased contribution of the LoRA parameters and the alignment induced by larger $s$ may explain our observation that higher $s$ leads to better performance. It's important to note that with a learning rate scheduler, the contribution of the second term would decay to zero.

### B.3 DERIVATION

Over-parameterization or linear-reparameterization in general has a non-trivial effect on the optimization dynamics. Here we analyze the update of the effective weight, to point out a rather surprising interaction between $s$ and $\eta$. Consider a standard update rule for SGD for $\mathbf{z}_{in} \in \mathbb{R}^{n \times 1}$, and $\mathbf{z}_{out} \in \mathbb{R}^{m \times 1}$ and $\mathbf{W} \in \mathbb{R}^{m \times n}$:

$$g(\mathbf{W}) = \frac{\partial\mathcal{L}}{\partial\mathbf{W}} = \frac{\partial\mathcal{L}}{\partial\mathbf{z}_{out}}\frac{\partial\mathbf{z}_{out}}{\partial\mathbf{W}} = \frac{\partial\mathcal{L}}{\partial\mathbf{z}_{out}}\mathbf{z}_{in}^T \tag{11}$$

We will denote $g(\mathbf{W}) = g$ from now on for clarity. For standard LoRA with parameters $\hat{\mathbf{W}} = \mathbf{W} + s\mathbf{BA}$, where $\mathbf{B} \in \mathbb{R}^{m \times r}$, $\mathbf{A} \in \mathbb{R}^{r \times n}$, the update rule on the effective weight is:

$$\hat{\mathbf{W}} \leftarrow \mathbf{W} + s\left(\mathbf{B} - \eta\frac{\partial\mathcal{L}}{\partial\mathbf{B}}\right)\left(\mathbf{A} - \eta\frac{\partial\mathcal{L}}{\partial\mathbf{A}}\right) \tag{12}$$

$$= \mathbf{W} + s\mathbf{BA} - s\eta\left(\left(\mathbf{B}\frac{\partial\mathcal{L}}{\partial\mathbf{A}} - \mathbf{A}\frac{\partial\mathcal{L}}{\partial\mathbf{B}}\right) + \eta\frac{\partial\mathcal{L}}{\partial\mathbf{B}}\frac{\partial\mathcal{L}}{\partial\mathbf{A}}\right) \tag{13}$$

We denote $g(\hat{\mathbf{W}})$ as $\hat{g}$. With the resulting effective update being:

$$\hat{g} = \left(\mathbf{B}\frac{\partial\mathcal{L}}{\partial\mathbf{A}} - \frac{\partial\mathcal{L}}{\partial\mathbf{B}}\mathbf{A}\right) - \eta\frac{\partial\mathcal{L}}{\partial\mathbf{B}}\frac{\partial\mathcal{L}}{\partial\mathbf{A}} \tag{14}$$

Computing the derivative for each variable introduces the dependency on $s$.

$$\frac{\partial\mathcal{L}}{\partial\mathbf{B}} = \frac{\partial\mathcal{L}}{\partial\mathbf{z}_{out}}\frac{\partial\mathbf{z}_{out}}{\partial\mathbf{z}_{res}}\frac{\partial\mathbf{z}_{res}}{\partial\mathbf{B}} = s\frac{\partial\mathcal{L}}{\partial\mathbf{z}_{out}}(\mathbf{A}\mathbf{z}_{in})^T \tag{15}$$

$$\frac{\partial\mathcal{L}}{\partial\mathbf{A}} = \frac{\partial\mathcal{L}}{\partial\mathbf{z}_{out}}\frac{\partial\mathbf{z}_{out}}{\partial\mathbf{z}_{res}}\frac{\partial\mathbf{z}_{res}}{\partial\mathbf{A}} = s\mathbf{B}^T\frac{\partial\mathcal{L}}{\partial\mathbf{z}_{out}}\mathbf{z}_{in}^T \tag{16}$$

Plugging it back in we have

$$\hat{g} = \left(\mathbf{B}\left(s\mathbf{B}^T\frac{\partial\mathcal{L}}{\partial\mathbf{z}_{out}}\mathbf{z}_{in}^T\right) - \left(s\frac{\partial\mathcal{L}}{\partial\mathbf{z}_{out}}(\mathbf{A}\mathbf{z}_{in})^T\right)\right)\mathbf{A} - \eta\left(s\frac{\partial\mathcal{L}}{\partial\mathbf{z}_{out}}(\mathbf{A}\mathbf{z}_{in})^T\right)\left(s\mathbf{B}^T\frac{\partial\mathcal{L}}{\partial\mathbf{z}_{out}}\mathbf{z}_{in}^T\right) \tag{17}$$

$$= s\left(\frac{\partial\mathcal{L}}{\partial\mathbf{z}_{out}}\mathbf{z}_{in}^T\mathbf{A}^T\mathbf{A} - \mathbf{BB}^T\frac{\partial\mathcal{L}}{\partial\mathbf{z}_{out}}\mathbf{z}_{in}^T\right) - s^2\eta\left(\frac{\partial\mathcal{L}}{\partial\mathbf{z}_{out}}\mathbf{z}_{in}^T\mathbf{A}^T\mathbf{B}^T\frac{\partial\mathcal{L}}{\partial\mathbf{z}_{out}}\mathbf{z}_{in}^T\right) \tag{18}$$

$$= s\left(\mathbf{BB}^T g - g\mathbf{A}^T\mathbf{A}\right) - s^2\eta\left(g\mathbf{A}^T\mathbf{B}^T g\right) \tag{19}$$

When $s$ is small both terms exist. When $s$ is large, the second term dominates. Since the last term is quadratic with $g$, one can safely ignore the second term when learning rate is sufficiently small. Simiarly, when using a learning rate scheduler, the contribution of the second term would decays to zero. The second-term can be interpreted as an alignment loss. Where the gradient is moves in the direction that aligns LoRA parameters.

## C  METHOD ILLUSTRATIONS

In our illustrations, we detail the distinctions between our method and other common strategies. Distributed Data Parallel (DDP) synchronizes the model at every iteration, with only the gradients being communicated between devices. This necessitates model synchronization across devices every iteration. Therefore, if there's significant delay in synchronization due to slow interconnect speeds or large model sizes, synchronization becomes a bottleneck. One way to mitigate this is through local optimization, often referred to as local steps or local SGD in federated learning. Here, instead of communicating gradients, model weights are shared. Local steps are known to converge on expectation, but they still require communicating the full model, which is loaded in half or full precision, which will quickly become infeasible in 1B+ size models

Our proposed method addresses both communication and memory issues by utilizing LoRA. Each device loads a unique set of LoRA parameters, and these parameters are updated locally. As discussed in our work, this enabled efficient exploration of full-rank updates. We communicate only the LoRA parameters, which can be set to be order of magnitude smaller than original model's size. Our approach balances single contiguous memory use with the ability to utilize more devices. The aim is to enable training of large models using low-memory devices.

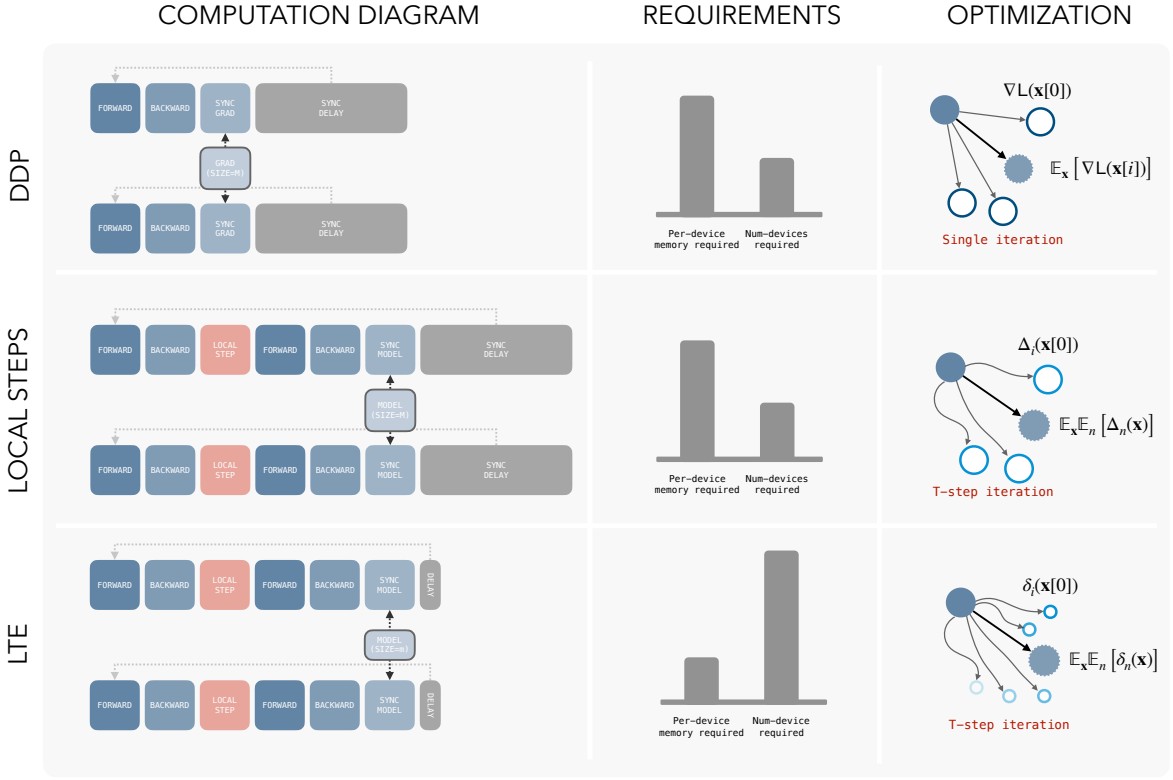

Figure 15: **Method illustration**: Comparisons between distributed learning methods to our method.

# D  ADDITIONAL RESULTS

In all of our experiments, we present two distinct curves for analysis: one that represents the total training data observed across all devices, and another that shows the training samples or tokens seen per device.

## D.1  VISION IMAGE CLASSIFICATION

We conduct additional experiments in image classification, covering datasets like CIFAR10 (Krizhevsky et al., 2009), CIFAR100 (Krizhevsky et al., 2009), STL10 (Coates et al., 2011), Caltech256 (Griffin et al., 2007), and SUN397 (Xiao et al., 2016). For these tests, we retuned all baseline learning rates. Detailed information about these datasets is available in Appendix A.1. For LTE we use rank of $r = 64$ and $N = 32$ heads.

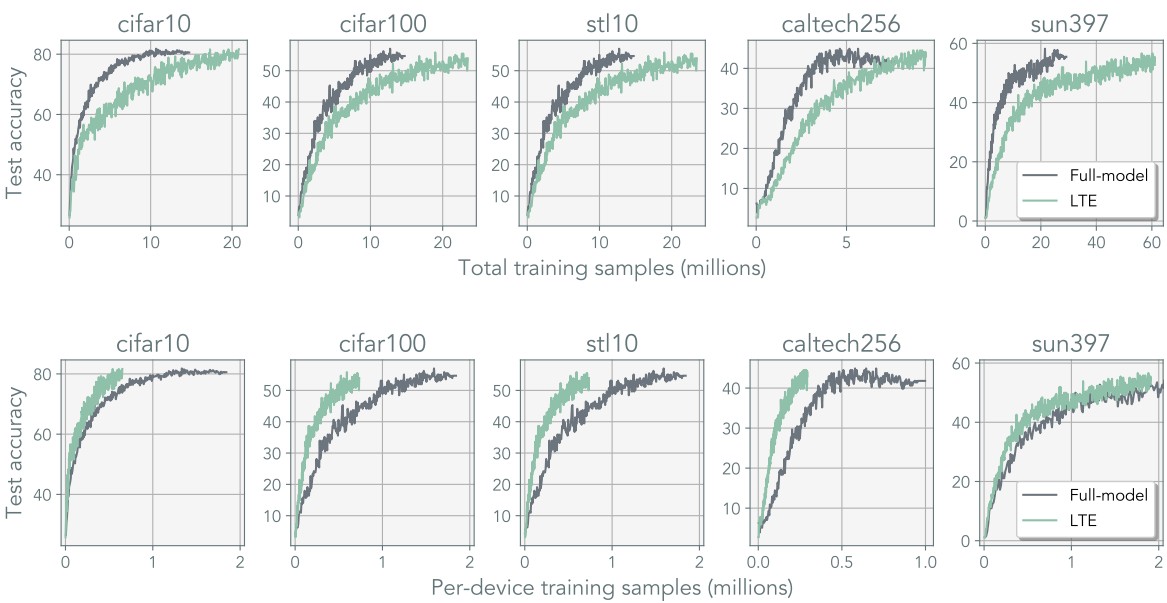

Figure 16: **Additional results on various image classification datasets using ViT-S**

### D.2 SCALING UP VIT MODEL SIZE

We train larger variants of the Vision Transformer (ViT) model. Details on these architectures are provided in Appendix A.1. Across all sizes, our results remained consistent. For ViT-L where we used a rank of $r = 128$.

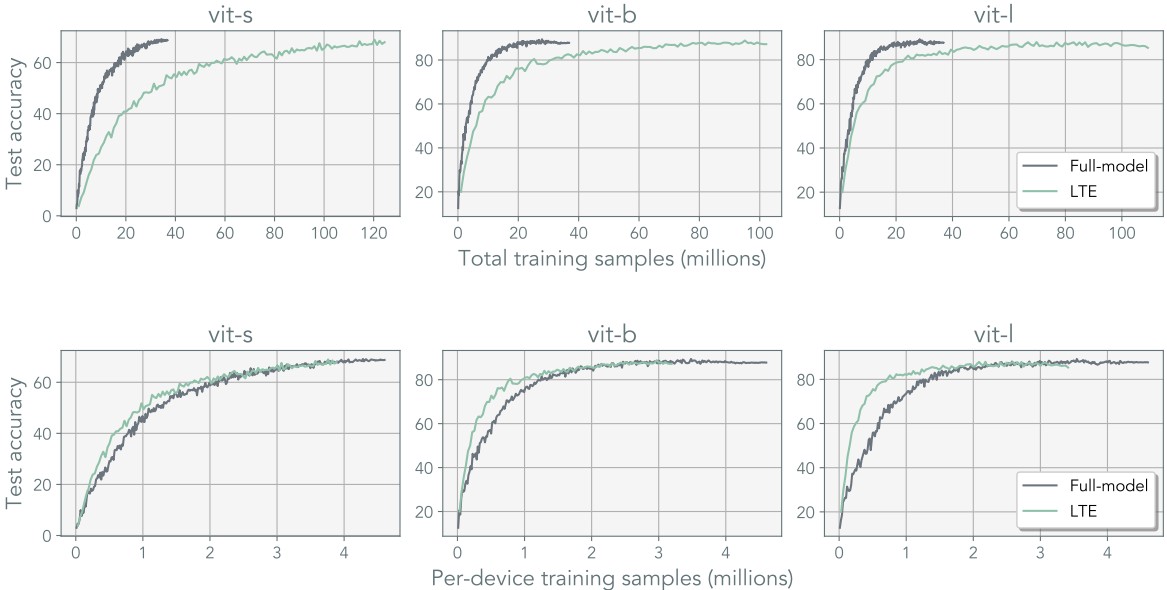

Figure 17: **ImageNet100 classification on varying ViT scale**

### D.3 LTE ON MLP-MIXER

To evaluate the generalizability of our method to non-Transformer based architectures, we train MLP-Mixer (Tolstikhin et al., 2021) using LTE. The specific details of the architecture used in datasets is listed in Appendix A.1. Our findings are consistent results across different scales of the MLP-Mixer. For Mixer-B, we use a rank of $r = 128$.

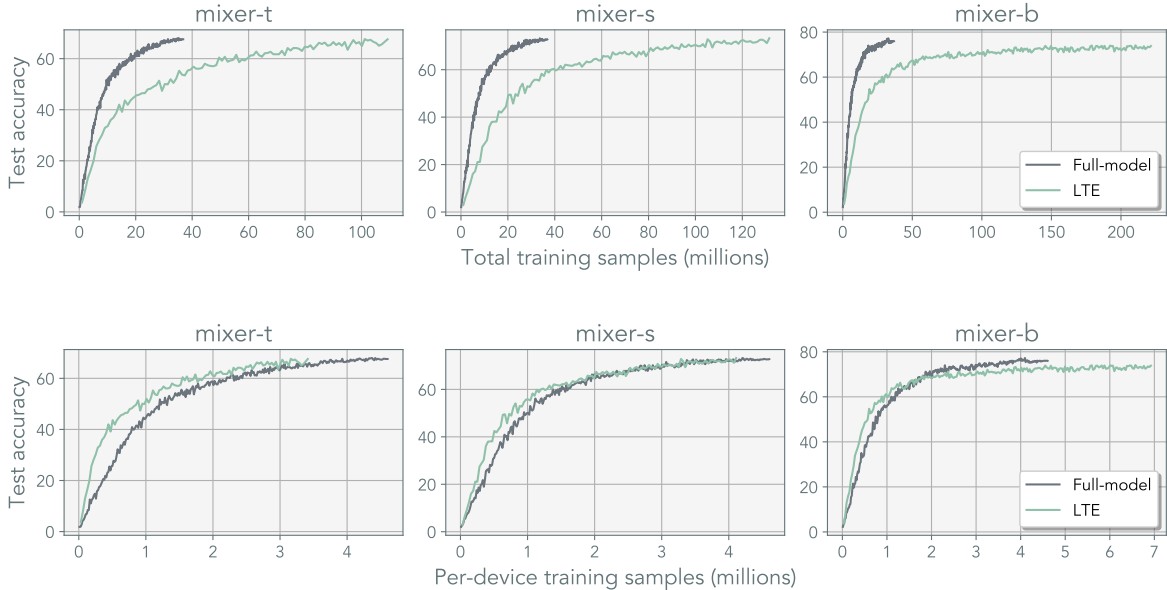

Figure 18: **ImageNet100 classification on MLP-Mixer of varying scale**

### D.4 LANGUAGE MODELING

We also apply our method to language modeling. For these experiments, we utilized the nanoGPT codebase (Karpathy, 2023). Detailed information about the architectures and datasets employed can be found in Appendix A.1. Shakespeare's dataset was trained using MiniGPT (Karpathy, 2023), while TinyStories (Eldan & Li, 2023) was trained on GPT2 (Radford et al., 2019). For Shakespeare, we used a configuration with rank $r = 16$ and $N = 32$ heads, and for TinyStories, we employed rank $r = 64$ and $N = 32$ heads. Consistent results were observed across all sizes. *Note*: The training for TinyStories is still in progress, and we will update the paper with the final results as soon as they become available.

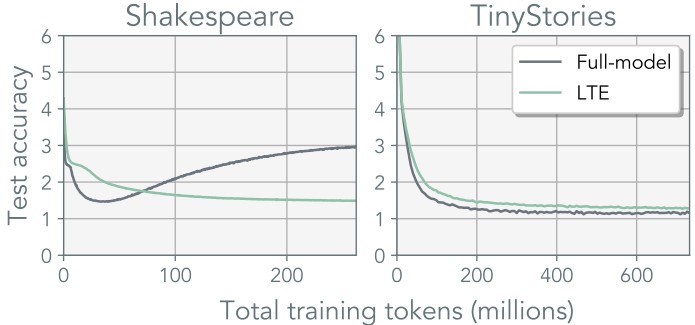

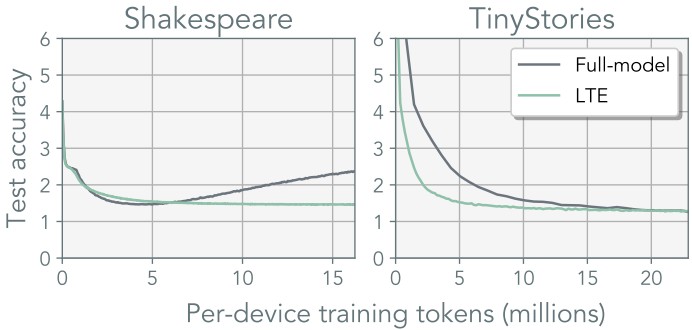

Figure 19: **ImageNet100 classification on MLP-Mixer of varying scale**

