# OpenReview forum: "Training Neural Networks from Scratch with Parallel Low-Rank Adapters"
_ICLR.cc/2024/Conference — Submitted to ICLR 2024_

### Official Review · Reviewer_q4qF · 2023-10-17

**Soundness:** 2 fair
**Presentation:** 3 good
**Contribution:** 2 fair
**Rating:** 3
**Confidence:** 4

**Summary:**

The paper introduces Lora-The-Explorer (LTE) a parallel low-rank training algorithm for multiple attention heads. The authors' main finding is that frequent synchronization, e.g. every batch, is unnecessary for effective training. Instead, stale gradients may be used effectively reducing the communication overhead. In an experimental evaluation the authors study the impact of number of heads, rank and synchronization delays on a Vision Transformer and ImageNet dataset. They observe that the overall number of necessary training iterations until convergence increases by roughly 20%, while requiring only a third of the memory.

**Strengths:**

- Straight-forward and logical idea to map LoRa to full trainings
- Significant improvement of trainable model sizes (factor of 3 claimed)
- Better initial training performance compared to the baseline
    * Food for thought: would it be meaningful to use LTE as a warm up approach to subsequently fine-tune with regular full-rank training?

**Weaknesses:**

- A 40% increase of necessary iterations is substantial and hinders practical use
- The authors present a thorough evaluation of LTE for a single dataset and a singular model architecture. Yet, it seems meaningful to provide additional experimental evidence for:
    * The stability of the finding for this particular use case, e.g. how large is the standard deviation for multiple different initializations/seeds?
    * How well does LTE generalize to other datasets and models architectures?

- Albeit a highly dense related work section, it seems to the reviewer that the authors have missed some closer related work in the realm of general stale data-parallel update schemes, e.g.:
    Coquelin, D., Debus, C., Götz, M., von der Lehr, F., Kahn, J., Siggel, M., & Streit, A. (2022). Accelerating neural network training with distributed asynchronous and selective optimization (DASO). Journal of Big Data, 9(1), 14.
    Chen, Y., Xie, C., Ma, M., Gu, J., Peng, Y., Lin, H., ... & Zhu, Y. (2022). SAPipe: Staleness-Aware Pipeline for Data Parallel DNN Training. Advances in Neural Information Processing Systems, 35, 17981-17993.

    It seems reasonable to mention and(!) possibly study/incorporate their findings

- In section 3.1, the authors mention that its assumed that all the parameters have the same initialization, but in section 4.1 you mention that LTE performs better when the heads are initialized to be different. It is unclear to the reviewer which scheme has been chosen for the experimentation.
- It is unclear how the weight synchronization is actually done. It is assumed by standard averaging like in DDP, but not explicitly stated especially with respect to the incorporation of stale gradients. How does the communication scheme for this look like (a form of master or an allreduce)? Where in Algorithm 1 is the synchronization happening?
- It would be meaningful to state the exact configuration of ViTs used for the full-rank training in Table 1
- The study is not reproducible as-is due to the source code not being released. If you intend to do so, it may be meaningful to hint it in the manuscript
- Fig. 7 should possibly also showcase the effects of quantization for a full rank training

Minor points:

- Figures and their textual reference are quite far apart
- Caption Fig 2., step (2), something is missing here, possibly the averaging and small delta
- Fig. 5 is actually a table and should be labeled as such

**Questions:**

There are a number of questions that the reviewer would like to learn more about and which should most likely be answered in the manuscript:

- How does your resource investment and utilization behave
    * Do you always commit exactly 8 GPUs/workers for all trainings? How do you deal with over-allocations of heads compared to workers
    * How does the wall-time behavior of LTE look like? Are your epoch execution time shorter or longer?
- It seem an apples-oranges comparison to allow larger models, i.e. with more heads, in the comparison against the baseline. The reviewer may have overlooked something, but why is this not the case?
- How would the rank r be practically determined? If the answer is a hyperparameter study, how detrimental is this for practical applications with an exponential growth of the hyperparameter search space?
- It is not mentioned how the non-attention heads in the ViTS network are trained. Is it a simple DDP-averaging with a frequency of 1?
- Although initialized to be orthogonal, the orthogonality of $A$ is not enforced or guaranteed, why would the cosine similarity be a
meaningful metric for it? ($A$ is a learned matrix will (likely) not be orthogonal anymore after several training steps). Is a general proof of update angles like in the following reference meaningful?

Frenkel, C., Lefebvre, M., & Bol, D. (2021). Learning without feedback: Fixed random learning signals allow for feedforward training of deep neural networks. Frontiers in neuroscience, 15, 629892.

---

> ### Author Response · Authors · 2023-11-18
> **Response to reviewer q4qF [1/4]**
>
> We thank the reviewer for the feedback and suggestions. We address the concerns below.
>
> Before addressing the points raised, we'd like to highlight that, as noted by the reviewer, one of our key findings is the effectiveness of using stale parameters. This is not the same as using stale gradients. While using stale parameters is one of our contributions to ensure practicality, this is just **one of several** contributions made in our work.
>
> Our paper provides many new insights, beyond the one mentioned by the reviewer:
> - A scientific investigation into whether LoRA style training can be used to train models from scratch (i.e., pre-training). All prior works rely on fine-tuning, and no work has yet shown this is possible. LoRA-style training has a low-memory footprint and can be used jointly with quantization.
> - We provide empirical evidence that the individual update iterates are high-rank (**Appendix B**), and then demonstrate that sequentially merging the LoRA parameter is insufficient to match pre-training performance (**Section 3**, and **Figure 4**).
> - We demonstrate that one can use parallel merging to approximate the update of multi-head LoRA, which we show empirically matches pre-training performance (**Section 3.1**).
> - We make a surprising observation that LoRA parameters do not converge to the same representation even if they were trained in parallel. We leverage this observation and show that we can train our model with infrequent synchronization. (**Section 4.2**, **Section A.4**)
> - We provide a thorough investigation on how varying the number of head, rank, and merge affects performance (**Section 4**).
> - Moreover, we provide additional mathematical intuition of LoRA dynamics and how the scaling factor s is not the same as scaling the learning rate. We provide the relationship standard SGD update and LoRA-parameterized update (**Section 3.3, Section B.1,Section B.2**)
>
> Beyond the list of contributions, the fact that it is possible to pre-train a deep model with LoRA alone should be of significant interest to the community.
>
> **> Practicality of our method**
> In response to the reviewer's concern about the 40% increase in necessary iterations affecting practicality: it's important to clarify that this increase, as stated in our paper, refers to the total data observed, not the data seen per device, which would better reflect the wall clock speed. In reality, each device observes less data. The trade-off of observing more data for parallelization is a fundamental assumption in federated learning and model averaging. Our framework, by enabling the pre-training of models on low-memory devices and in environments with low interconnect speeds, ventures into a previously unexplored research area.
>
> To re-emphasize, our work presents a first scientific investigation of using adapters alone to pre-train deep models from scratch. We expect the efficiency of this methodology to improve with future revisions.
>
> **> Singular architecture**
> Originally, our focus was on transformers, given their prominence in training models across various modalities. However, in response to the suggestion for broader experimentation, we have expanded our scope. This includes additional experiments with:
> - ViT models of various scales, including ViT-S, ViT-B, and ViT-L (detailed in **Appendix A.3**).
> - Non-transformer based architectures, specifically the MLP-Mixer (outlined in **Appendix A.4**).
> - Comprehensive vision tasks using ViT on datasets such as CIFAR10, CIFAR100, STL10, CALTECH256, and SUN397 (as described in **Appendix A.2**).
> - Language modeling experiments on Shakespeare and TinyStories on miniGPT and GPT2 respectively (**Appendix D.4**).

---

> ### Author Response · Authors · 2023-11-18
> **Response to reviewer q4qF [2/4]**
>
> **> Comparison to existing data-parallel updates schemes**
> The reviewer states the similarity of our work to literatures in federated learning. However we would like to emphasize that our work extends beyond methodologies explored in federated learning. Federated learning, including the works mentioned by the reviewer, focuses on data-parallelism where each node has to communicate either the gradient or the parameters of the full-model.  As models become bigger than 10M, it quickly becomes communication bound. This serves as the main bottleneck for scaling up these algorithms in standard federated learning. In contrast to these works, our method **combines both data-parallelism and model-parallelism**. Our method utilizes unique low-rank exploration by each device. Even though the individual updates are low-rank, the collective average across the devices materializes in to a full-rank update. Because our method only communicates the low-rank updates, it can scale up more efficiently to larger models.
>
> We acknowledge that using stale estimates is not a novel concept, and we **do not** claim it as such. The referenced works are indeed relevant to our study, and we have **cited them in the revision**. We highlight the major differences below:
>
> 1. DASO demonstrates hierarchical communication protocol with delayed synchronization. Unlike our method, DASO, requires the full model to be communicated every T iterations. The key distinction is that our approach does not depend on the full model for training. By utilizing LoRA, we significantly reduce communication overhead. This means we transmit only a fraction of the parameters compared to what would be required otherwise. Additionally, our method extends QLoRA benefits, allowing training of models that previously couldn't be loaded in full precision.
> 2. The main contribution of SAPipe centers around handling stale gradients, with their proposed methods: "delay compensation" and "weight prediction". The work of SAPipe focuses on improving the use of stale gradient estimates. Here we highlight that infrequent synchronization reduces communication frequency but **not** bandwidth. The focus of work is to reduce both communication frequency and bandwidth. We demonstrate that this is possible with parallel low-rank updates using LoRA.
>
> We recognize the relevance of these works, incorporating their specific results and methodologies is outside the scope of our original investigation: "is it possible to pre-train a model from scratch using low-rank adapters alone". We have referenced these works in **Section 5**. We have also added additional illustrations in **Appendix C** to highlight the differences to prior methods.
>
> **> Clarification to Section 3.1**
> Section 3.1 and 4.1 are consistent, and we apologize if our wording caused any confusion. Each head is always initialized differently. The original text in Section 3.1 was to state that both sides of the equations $B_n = \hat B_n$, $A_n = \hat A_n$  are initialized with the same values. We did not mean to say that each head was initialized with the same distribution. Each LoRA head is initialized independently of each other and are not the same between each other. **Appendix A.4** highlights the importance of the parameters being initialized differently. We **revised Section 3.1** to improve clarity.
>
> **> Weight synchronization**
> As detailed in **Section 4.5** we experimented with 2 implementations for weight synchronization:
>
> 1. Using standard DDP.
> 2. Using parameter server.
>
> When using DDP style communication, only the LoRA parameters are communicated. Each node broadcasts the LoRA parameters using all-reduce. The communication would require roughly $N^2$ communication, and bandwidth required is orders of magnitude smaller compared to full-model training. In our implementation, we simulate multiple nodes with a single GPU by using PyTorch vmap. We use this implementation for our experiments.
>
> Using a parameter server would require approximate N communication to the server with only the delta weights, and N communication of the quantized weights back to the nodes.
> The advantage of a parameter server would scale if the quantized weights can be efficiently communicated back to the nodes. We did not explore improving this setup.
>
> The reviewer further states that we do not explicitly state “the incorporation of stale gradients”.  We would like to **provide clarification** to this statement. The gradients we compute are not stale; the staleness comes from taking multiple optimization steps before merging. Updates at each device/head are exact. The “staleness” mentioned in the paper is a direct consequence of merging every T iterations. The LoRA parameters themselves are the updates for the main weight. Algorithm 1 and Figure 3 is exactly how the algorithm is implemented. Let us know, if further clarification is needed.

---

> ### Author Response · Authors · 2023-11-18
> **Response to reviewer q4qF [3/4]**
>
> **> Figure 7 should showcase quantization**
> We have **updated Figure 7** to include both scenarios: with and without quantization. The inclusion of quantization requires more memory but is 20% faster. Conversely, the benefits of omitting quantization become more pronounced in larger-sized models, where the optimizer states consume a significantly larger portion of memory compared to the model itself
>
> **> Resource investment and utilization**
> The resource allocation for our experiments varied, utilizing either 4 or 8 GPU devices, depending on availability at that time. As outlined in the paper, we simulate multiple heads within the same GPU using vmap, underscoring that our focus is not on systems design but on methodology.
> For our experiments, we simulate having 16-64 devices within just 4-8 GPUs, making a direct comparison of wall clock time challenging. We provide theoretical utilization to provide some perspective:
>
> Without custom kernels, the LTE forward/backward pass is about 66% slower, but the parallel convergence time is significantly quicker—upto 65.7% faster. This means that the overall wall clock time could be easily made comparable when using many low memory-compute devices.
>
> Our method particularly excels in a communication-bandlimited regime. If, for example, the cost of communicating the gradient of the full model is comparable to the time taken for a forward/backward pass, then the communication of LTE parameters would be much less, making the communication cost relatively negligible. In such a scenario, our method could train approximately 20% faster. This advantage becomes more pronounced with increased parallel computation, assuming our trends continue with more LoRA heads.
>
> Long-term, what our method enables is the use of older, less powerful GPU devices to train large models. If utilizing low-memory, slower devices proves to be significantly more cost-effective than current high-end accelerators, it might be worthwhile to reallocate resources towards these older devices. This paper sets up the groundwork for this direction.
>
> The runtime for these experiments was assessed using a ViT-S model with a batch size of 1024 on V100 GPUs.
>
> **> Determining rank r**
> In our experiments, we chose the rank r and the number of heads N so that r x N is approximately the maximum dimension of the Vision Transformer (ViT) models. The ideal selection of r x N should vary depending on the task, with simpler tasks potentially requiring a smaller r. Additionally, the choice of r is influenced by the maximum memory capacity available on the device. In our approach, we do not dynamically adjust r. As shown in Table 1, we observed that under a fixed training epoch, performance drops significantly at lower ranks, but as the rank increases, performance reaches a peak and then levels off.
>
> Exploring the optimal r is an intriguing question we are currently investigating for future works. Some potential directions are:
> 1. Estimating the rank of the deltas to use as a proxy for saturation,
> 2. Performing a single forward-backward pass on a larger model in the main server to estimate the gradient rank, and
> 3. Applying techniques similar to those used in critical batch-size and batch-noise studies [1][2] to observe the saturation of gradient ranks when adding a new device. These methods have proven to be efficient for computation.
>
> We plan to include these experiments in the follow up work.
>
> **> Expansion beyond linear layers**
> Indeed, our initial work did not explicitly address this aspect. Regarding vector-weight layers like layer-norm affine parameters, we experimented with various approaches, but none showed significant impact. These approaches included:
> - Training one model to handle the layer-norm layer.
> - Simulating LoRA-like parameterization for vector-based layers, with configurations such as A=vector, B=vector, or A=vector, B=scalar.
> Simple averaging of parameters.
> - Opting not to train these layers at all.
>
> Surprisingly, all these methods yielded similar results. We have now included these details in the Appendix A.1.. As for other layers, such as convolutional layers, they can also adopt LoRA-style parameterization where layer B is a 1x1 convolution. For embedding layers, we opted for averaging. These layers are synchronized when the LoRA parameters are synchronized. We have **included these details in Appendix A.1**.
>
> **> Orthogonality of A**
> The LoRA heads act as directional components in the updates, similar to derivatives. Hence, cosine similarity is a natural metric for measuring direction similarities across each head. As illustrated in our experiments, the surprising findings is that the LoRA heads do not seem to converge to the same representation, as discussed in Section 4.2.
>
> ---
> [1] McCandlish et al. “An Empirical Model of Large-Batch Training”
> [2] Kaplan et al. “Scaling Laws for Neural Language Models”

---

> ### Author Response · Authors · 2023-11-20
> **Response to reviewer q4qF [4/4]**
>
> **> Architecture details**
> As ViTs are a popular family of architecture we originally omitted the exact detail. We have now included them in the **Appendix A.1**.
>
> **> Code release**
> Code to replicate our results will be released at the end of the reviewing cycle irrespective of acceptance decision. We have **added** the promise of the code release into the introduction.
>
> **> Minor points: figure location and typos**
> Figure 1 originally had 2 figures, where the second figure was referenced later in the paper. We agree the textual reference is far and we have split the figure into 2. The original right-side of Figure 1 is now Figure 4. We have **fixed** the typo if Figure 2 and Figure 5 is now **fixed** and is Table 1.

---

> > ### Author Response · Authors · 2023-11-22
> > **Additional orthogonality experiments**
> >
> > In our original work we measured cosine distance between the heads as an indicator of orthogonality. We have repeated the experiments using Grassman distance of the LoRA heads throughout optimization and observed their distances to remain consistent throughout optimization. See **Appendix A.5 Grassman and Cosine distances of LoRA heads.**

---

### Official Review · Reviewer_i1ro · 2023-11-01

**Soundness:** 2 fair
**Presentation:** 2 fair
**Contribution:** 2 fair
**Rating:** 5
**Confidence:** 4

**Summary:**

The paper explores the extension of Low-Rank Adaptation (LoRA, Hu et al., 2021) for pre-training deep learning models. The authors propose LoRA-the-Explorer (LTE) that facilitates parallel low-rank updates/heads via infrequent synchronization across multiple compute nodes. LTE bridges the performance gap induced by a single LoRA head and recovers the standard model training from scratch. The experiments are conducted for Vision Transformer models using ImageNet100 to demonstrate the competitiveness of the proposed LTE with standard Distributed Data-Parallel (DDP) on 8 GPUs. At the cost of an additional 40% longer training time, LTE with 32 parallel LoRA heads (rank =64) converges to the same top-1 accuracy for ViT-S model.

**Strengths:**

This work boldly attempts to leverage the highly popular low-rank fine-tuning method (LoRA) for pre-training models from scratch. The paper carefully studies the performance degradation by drop-in LoRA, acknowledges the limitations, and offers future opportunities for the community to explore this line of research. The central idea of the proposed work is to approximate full-rank weight as a linear combination of low-rank weights, termed multi-head LoRA (MHLoRA). Section 3.1 is easy to understand and shows that single-head LoRA is equivalent to merging the multi-head LoRA at every iteration (tight synchronization). To alleviate this bottleneck of frequent synchronizations, authors propose periodic merging.

**Weaknesses:**

1. It will be worthwhile to examine, compare, and contrast a parallel body of work ReLoRA (Stack More Layers Differently: High-Rank Training Through Low-Rank Updates, Lialin et al., arXiv, July 2023) that presents a similar core idea.

2. Section 1 presentation can be further improved. The main findings and contributions in the middle seem to break the flow and could be considered a closing paragraph. Moreover, some of the items in these findings are the organization of the paper (eg, the last point on related work). It would be helpful to condense and limit it to highlight the key contributions and novelty of the work.

3. The key takeaways of the work are in the experiment sections. It would be helpful to highlight those and map them to theoretical insights.

4. The use of deep learning/neural networks in the title and body must be carefully examined as it may be misleading in its current form. The technique is presented for Transformer models, specifically ViTs. The authors should consider using ViTs instead to align the readers' expectations to the presented work.

5. It would be helpful to illustrate the parallelism and mapping to compute nodes in the proposed technique:
 parallel low-rank adapters corresponding to multi-head LoRA (MHLoRA) and multiple heads in Transformer models that allow natural parallelism.

6. The Figure captions should specify the model (ViT-S?) being used.

7. Page 3: Last 2 lines:
- Is Denominator N missing from the summation term? ….*when either (summation)  is equal to (single LoRA head) $B_nA_n$*…..
- What does j represent in $n \neq j$ ?

8. Page 5: repetitive use  ..*learning rate of*…

9. Page 8: define $N_{ddp}$ and $N_{lte}$

**Questions:**

Please refer to the Weakness section.

---

> ### Author Response · Authors · 2023-11-18
> **Response to Reviewer i1ro [1/1]**
>
> We thank the reviewer for the comments. We address the concerns below.
>
> **> Section 1 presentation**
> Thank you for the suggestion. We have **revised Section 1** and condensed the contribution list as suggested.
>
> **> Additional experiments**
> To further validate the effectiveness of our method, we have expanded our experimental scope beyond ViT-S on ImageNet100/ImageNet1k. Let us know if there are additional experiments you would like to see.
>
> - We've added five new vision datasets: CIFAR10, CIFAR100, STL10, CALTECH256, and SUN397 (**Section D.1**).
> - We've conducted tests on various ViT models, including ViT-S, ViT-B, and ViT-L (**Section D.2**).
> - We have also extended our experiments to non-transformer based architectures including: MLP Mixer-T/S/B (**Section D.3**).
> - We have also added language modeling experiments: Shakespeare on miniGPT, and TinyStories on GPT2 (**Section D.4**).
>
> **> Illustration of model/data parallelism**
> To provide a clearer understanding of the parallelism of various methods, we have **added diagrams in Appendix C**. We added illustrations for computation diagram, memory requirements, and visual optimization strategies between DDP, local steps, and our method LTE. It's important to note that MHLoRA, which we use as a baseline, requires full forward and backward synchronization. Therefore, it operates within the standard frameworks of DDP or/and local steps/SGD.
>
>
> **> Comparison to ReLoRA**
> ReLoRA is a concurrent work that sequentially merges a single LoRA head. We argued that this style of training is insufficient to pre-train the model from scratch. As expected from our results and arguments, the authors of ReLoRA do not demonstrate that it is possible to pre-train models from scratch using their method. Moreover, one of our baseline resembles ReLoRA, and we have already acknowledged this in our original submission. This is referred to as “LoRA with Merge” as shown in **Figure 4**. We further outline the differences below:
>
> 1. Similar to our baseline, ReLoRA trains a single LoRA which is repeatedly merged and reset.
> 2. Our findings demonstrate that “LoRA with merge”/ReLoRA fails to match the performance of full-model pre-training (**Figure 4**).
> 3. We provide empirical evidence that the update iterates of transformers are high-rank, which we know a single LoRA cannot express. (**Appendix B**)
> 4. In contrast to ReLoRA, we demonstrate that parallel low-rank updates can successfully recover pre-training performance by approximating high-rank updates.
> 5. ReLoRA does not pre-train models from scratch. Instead ReLoRA rely on full model training for majority of the loss, and then switches over to LoRA-based training. As depicted in ReLoRA's Figure 1, the method uses full-model training from initial loss of 5.0+ to 3.6 and then switches over to LoRA to further reduce the loss from 3.6 to 3.4. Hence ReLoRA does not reduce the original computational requirement.
> 6. Lastly, we would like to point out that this work remains unpublished and was archived close to the ICLR submission deadline.
>
> If there is something that we have mistaken, please let us know.
>
> **> Caption suggestions, typos, and definitions (6-9)**
> We have revised our figure captions to specify the architecture and dataset used in each case. Additionally, we have corrected the typos throughout the manuscript and provided definitions for previously undefined variables. We also addressed a typo in one of the equations and clarified the meaning of the summation terms, which are intended to represent the sum of all LoRA heads excluding head $n$.

---

### Official Review · Reviewer_aA3h · 2023-11-01

**Soundness:** 2 fair
**Presentation:** 3 good
**Contribution:** 3 good
**Rating:** 6
**Confidence:** 3

**Summary:**

The paper extends the application of low-rank adapters from the domain of fine-tuning to model pre-training. The central innovation is the employment of a linear combination of low-rank adapters used in parallel for training models from scratch. The idea has the potential of enabling memory-efficient and communication-efficient model pre-training.

**Strengths:**

1. The paper delves into an underexplored area by investigating the potential of parallel low-rank updates for memory-efficient and communication-efficient model pre-training. This is highly relevant in the context of contemporary computational constraints.

2. The introduction of multi-head low-rank adapters that integrate into model parameters constitutes a novel contribution to the field. This idea could generalize to multiple training paradigms, thereby adding considerable value to existing research.

**Weaknesses:**

1. The paper acknowledges its own limitation as a proof-of-concept work. Although the idea is compelling, there is insufficient evidence to support its feasibility for large models or complex tasks.

2. The manuscript would benefit from an in-depth theoretical analysis that substantiates the proposed approach, thereby addressing its current shortcomings.

**Questions:**

While the extension of Low-Rank Adaptation (LoRA) to model pre-training is undoubtedly an interesting avenue, the paper does not convincingly address the efficacy of using a combination of low-rank adapters in this context. More empirical and theoretical work are needed to validate this approach.

While the impact on resource utilization is stated, quantifying or benchmarking the reduction in memory or communication data usage would be beneficial.

---

> ### Author Response · Authors · 2023-11-18
> **Response to Reviewer aA3h [1/1]**
>
> We thank the reviewer for the suggestions.
>
> **>Additional experiments**
> To demonstrate the generalizability of our method, we have conducted additional experiments, which are now included in the revised manuscript:
> - We have conducted experiments with various sizes of ViT models on ImageNet100 (**Section D.2**).
> - Our method, LTE, has been applied to a non-transformer architecture, specifically the MLP-mixer (**Section D.3**).
> - Additional datasets have been incorporated into our testing, including CIFAR10, CIFAR100, STL10, CALTECH256, and SUN397 (**Section D.1**).
> - Additional experiments on language modeling tasks including Shakespeare, and TinyStories on miniGPT and GPT2 respectively (**Section D.4**).
>
> Across all these experiments, we have observed results consistent with those reported in our original findings.
>
> **> Theoretical analysis**
>
> Understanding how existing theoretical frameworks fit in with our findings is undoubtedly interesting. While we defer a comprehensive theoretical investigation to future works, we identify several relevant works that align with our research:
>
> 1. Understand optimization when using unsynced parameters.
> 2. The science of model averaging and merging.
> 3. Understanding the expressivity of the LoRA.
>
> Investigating optimization with unsynchronized parameters is well-established in the field of federated learning [1]. This topic has been explored in depth, with studies on the convergence guarantees of local SGD on heterogeneous data [2].
>
> The area of model averaging and merging is an ongoing research topic within the Linear Mode Connectivity (LMC) and model-averaging communities. Current studies investigate optimal merging and try to understand why large models have better merging capabilities [3][4][5]. Unifying these 2 subfields would be an interesting direction moving forward.
>
> Recently, there have been efforts to estimate the expressivity of LoRA [6]. Understanding the types of functions LoRA can model necessitates a consideration of over-parameterization literature [7][8][9] and studies on dimensional bottlenecks [10].
>
> While we acknowledge that extending the theoretical analysis would indeed be insightful, we believe that such an extension is beyond the scope of our current submission. We would be happy to further the discussion in this front. Did the reviewer have any specific theoretical analysis they wanted to see or be discussed?
>
>
> ---
> [1] Stitch et al., "Local SGD Converges Fast and Communicates Little".
> [2] Li et al., "On the Convergence of FedAvg on Non-IID Data".
> [3] Zhou et al., "Lima: Less Is More for Alignment".
> [4] Wortsman et al., "Model Soups: Averaging Weights of Multiple Fine-Tuned Models Improves Accuracy Without Increasing Inference Time".
> [5] Ortiz-Jimenez et al., "Task Arithmetic in the Tangent Space: Improved Editing of Pre-Trained Models".
> [6] Zeng et al., "The Expressive Power of Low-Rank Adaptation".
> [7] Arora et al., "Implicit Regularization in Deep Matrix Factorization".
> [8] Valle-Peréz, "Deep Learning Generalizes Because the Parameter-Function Map is Biased Towards Simple Functions".
> [9] Huh et al., "The Low-Rank Simplicity Bias in Deep Networks".
> [10] Yang et al., "A Spectral Condition for Feature Learning".

---

### Official Review · Reviewer_Sixn · 2023-11-05

**Soundness:** 1 poor
**Presentation:** 1 poor
**Contribution:** 2 fair
**Rating:** 3
**Confidence:** 4

**Summary:**

The paper introduces the use of LoRA for pretraining in the context of neural networks. LoRA, originally designed to approximate weight updates as a matrix by decomposing it into two low-rank matrices, is typically employed after the initial pretraining phase. In this paper, a novel approach is proposed wherein the update matrix is represented as a mixture of low-rank matrices referred to as "heads." These heads collectively approximate the updates to the original weight matrix. Interestingly, various workers operating on potentially distinct GPUs compute updates to the LoRA matrices, and these updates are eventually used to update the original weight matrix after several iterations. This innovative method seeks to improve the pretraining process in neural network training.

**Strengths:**

***Federated Learning-Inspired Approach*** - The paper introduces a novel algorithm aimed at updating the initial parameters following the independent training of parallel LoRAs for several iterations.

***Innovative Concept*** - The concept introduced is quite refreshing, departing from the conventional practice of directly updating the original parameters and instead approximating them from a combination of low-rank matrices. This approach bears similarities to the concept of a "mixture of experts," as explored in other research papers such as "AdaMix: Mixture-of-Adaptations for Parameter-efficient Model Tuning." Notably, the authors apply this method to the pretraining phase, in contrast to the more common practice of fine-tuning the model

**Weaknesses:**

***Insufficient Experimentation*** - The paper lacks comprehensive experimentation, as it fails to include a comparison with competing methods or an initial set of experiments to validate the effectiveness of their proposed approach. For instance, the absence of comparisons to full fine-tuning or the use of a single LoRA with a pre-trained model in a conventional context is notable. The inclusion of more experiments would significantly enhance the paper's credibility.

***Lack of Elaboration*** - The paper also suffers from a lack of necessary details. Crucial information regarding the dataset employed in the experiments is conspicuously absent. Furthermore, the figures provided do not effectively convey substantial information. To address this issue, a solution might be to relocate Figures 2 and 3 to an appendix, thereby creating additional space for a more detailed explanation.

***Enhanced Clarity in Writing*** - The motivation behind incorporating LoRA during the pre-training phase remains unclear throughout the paper. The initial paragraphs fail to provide a concise introduction to the fundamental concepts explored in the paper, exemplified by the use of the term "LoRA head" in the introduction without prior definition. Additionally, the authors' use of "merging" to describe the combination of LoRA weights with W during training is not adequately clarified until a reader has delved deep into the paper. Improving these aspects would enhance the overall clarity of the paper.

**Questions:**

- ***Synchronization of Updated Weights*** - You mention that the original model's weights are updated by merging LoRA weights after several iterations. However, it's not clear whether this updated information is effectively communicated to all the workers involved in the process.
- ***Clarification of Experimental Particulars*** - The paper lacks critical information regarding the dataset used for training the models. The absence of details concerning the dataset used in the experiments is a notable gap in the presentation.
- ***Consideration of Alternative Datasets*** - While the authors mention testing their methods on the Imagenet dataset, it would be beneficial to understand whether they have considered using other datasets for comparison and analysis.
- ***Interpreting Figure 1*** - Figure 1 appears to suggest that only the LoRA matrices are updated without any mention of a "merging" process. This leaves room for confusion, as it might imply that the original weight matrix W is randomly initialized, potentially impacting training stability. It would be valuable to clarify whether this is indeed the case.

---

> ### Author Response · Authors · 2023-11-18
> **Response to Reviewer Sixn [1/2]**
>
> We appreciate the reviewer's feedback and address their concerns below. The reviewer acknowledges the novelty and innovation of our method but does not elaborate on their reasons for rejecting the paper. Many of the reviewer’s concerns were already addressed in the original submission. We are open to adding specific experiments or clarifications if requested.
>
> **> Insufficient Experimentation:**
> Could the reviewer provide clarification to what they meant by insufficient experimentation? Given the set of experiments we have run, what experiments would you like to see more?
>
> To the best of our knowledge, the use of low rank adapters alone to train models from **scratch** is an unexplored area of research. We performed thorough comparisons against all reasonable baselines. These comparisons include:
>
> - (1) training with an adapter alone,
> - (2) training a model from scratch, and
> - (3) training a model with merges.
>
> These experiments are shown in **Figure 1, Figure 4, Table 1, and Figure 12**.
>
> **> Similarities to AdaMix**
> The reviewer mentions similarities of our work to AdaMix, which focuses on averaging MoEs for model fine-tuning, and we acknowledge its relevance. However, our method differs in many ways:
>
> 1. MoE splits each FFN of a model block (e.g., Transformer block) into different devices. Which requires frequent communication within a single forward pass. This is a crucial aspect that differentiates our work from AdaMix. AdaMix would require information to be **routed and gathered at every layer**, thereby increasing communication costs significantly. Our work demonstrates that it is possible to use parallel low-rank updates that require infrequent synchronization and low communication bandwidth.
> 2. MoE-style training operates under a different setting, where the communication is assumed to be negligible (servers with specialized communication hardware). We are interested in communication limited settings.
> 3. AdaMix only operates on the FFN layer of the attention block, and does not extend to other parts of the layers. Our method can train on all linear layers.
> 4. Our work focuses on pre-training, whereas the work of AdaMix focuses on fine-tuning.
>
> We have **updated Section 5** to cite AdaMix as related works.
>
>
> **> Absence of comparison to full-finetuning or the use of single LoRA**
> We would like to emphasize that **this statement is incorrect**. Contrary to the statement, we have compared our method against both full-finetuning and single LoRA in several experiments:
>
> - Figure 1 (Figure 1 left from previous revision) compares against full model fine-tuning with single LoRA.
> - Figure 4 (Figure 1 right from previous revision) compares LoRA with merging with full model fine-tuning.
> - Table 1 compares the baseline of full-fine tuning against our method.
> - Figure 6 compares LTE (our method) with single LoRA with merge (LTE 1 head).
> - Figure 8 compares single LoRA, full-finetuning, and LTE on least-squares
>
> **> Lack of Elaboration / Clarification of Experimental Particulars**
> We **refute** the reviewer's claim that we are withholding dataset and training details. Our methods utilize well-known datasets like ImageNet100 and ImageNet1K on the widely used ViT architecture, adhering to standard PyTorch vision training protocols:
> - https://pytorch.org/blog/how-to-train-state-of-the-art-models-using-torchvision-latest-primitives/
> - https://github.com/pytorch/vision/tree/main/references/classification
>
> We have already appropriately referenced the details in our submission. See:
> - Introduction: Principal findings and contributions:
> - Section 4.1, second paragraph.
> - Section 3.3 Implementation details.
> - Appendix A.1 Training details.
>
> For further clarity, ImageNet1K is a collection of 1000 classes with approximately 1.3M images, and ImageNet100 is a 100-class subset with the same resolution. We decided to explicitly include these details in **Appendix A.1**
>
> **> Consideration of Alternative Datasets:**
> Thank you for the suggestion, we have ran additional experiments:
> - We have included 5 new vision datasets: CIFAR10, CIFAR100, STL10, CALTECH256, SUN397
> (**Section D.1**)
> - We have also ran on variations of ViT including: ViT-S, ViT-B, ViT-L
> (**Section D.2**)
> - We have also extended our experiments to non-transformer based architectures including: MLP Mixer-S
> (**Section D.3**)
> - We have added language modeling tasks on Shakespeare and TinyStories (**Section D.4**)

---

> ### Author Response · Authors · 2023-11-18
> **Response to Reviewer Sixn [2/2]**
>
> **> Enhanced Clarity in Writing**
> The reviewer questions the clarity of our motivation behind using LoRA in the pre-training phase. While, we have already discussed this in
>
> - Section 1. Introduction
> - Section 2.1 preliminary on LoRA
> - Section 6. Conclusion.
>
> We highlight our motivation again:
> - LoRA has become the community favorite for fine-tuning models due to low memory footprint, and its compatibility with quantized weights.
> - Hence, to extend the benefits of LoRA, we investigate whether it is possible to pre-train a model using LoRA alone.
> - Our method sets up the groundwork for what was not possible before, training large models on low-memory devices. This is especially important where there is poor interconnect speed, and cross-node model-parallelism is not a practical option. It is well known in the community that communication is the main bottleneck for efficient distributed training, even within the same cluster system. To the best of work knowledge, this is a new direction that has not been thoroughly explored in the community.
>
> We have revised and shortened our introduction. (**See Section 1.**)
>
> **> Weight Synchronization:**
> The synchronization of weights occurs at every T iteration, this is indicated in several places in our paper:
>
> - Algorithm 1. Noted as “Merge LoRA params”
> - Figure 3. “We parameterize the model with multiple LoRA heads and train them independently for $T$ iterations using different mini-batches sampled from the same distribution.”
> - Section 3.2, “For each LoRA head $n$, the parameters are optimized with respect to its own data partition for $T$ iterations resulting in an update ... We do not synchronize the optimizer state across workers. After the optimization, the resulting LoRA parameters are synchronized to compute the final update for the main weight...”
>
> We have included further illustrations in **Appendix C** , added a comment in **Algorithm 1**, and added communication details in **Appendix A.1**.
>
> **> Interpreting Figure 1:**
> There may have been a mis-interpretation of Figure 1. by the reviewer.
> We do not apply merging in Figure 1 (left) (Figure 1 in new revision), instead this is a comparison between standard training with single LoRA as denoted in section 3 and also in the caption.
> Figure 1 (right) (Figure 4 in new revision) is the comparison with merge and includes a "Merge" label in the legend, and the caption states that comparison is between LoRA with and without merging.
> We have split the figure into two separate figures (Figure 1 and Figure 4 in the revised manuscript) and **revised the caption for Figure 1 and Figure 4** for clarity.

---

### Author Response · Authors · 2023-11-19
**Response reviewers**

We thank all the reviewers for dedicating their time to review our work.

We have made efforts to address all the concerns raised by each reviewer. Direct revisions have been made to the manuscript, and we encourage the reviewers to examine the updated draft.

For ease of reference, any minor text edits are marked in red. Additionally, we have colored the titles of sections that have been rewritten or newly added in red. We will ensure the text fits the page limit at the end of the review cycle.

Let us know if there are any further questions, and experimental demands. We hope to have a fruitful discussion.

---

### Meta-Review · Area_Chair_Bdkg · 2023-12-09

**Metareview:**

This paper proposes the use of LoRA adapters for from-scratch pretraining. The motivation of using LoRA adapters is primarily to reduce communication costs, but also to decrease training costs on each worker (e.g. by quantizing the main model as in QLoRA). The resulting setting is essentially federated learning, and the paper shows that LoRA updates can be used for from-scratch training with a 40% increase in cumulative training samples (which is relatively modest when considering the gains from parallelization). As the authors state, this work is primarily a proof-of-concept. However, it touches on an incredibly common setting where there is a huge amount of prior work that is only mentioned towards the end of the paper (rather than being compared to directly). Given that the goal is to reduce communication costs without harming convergence speed, the method should be compared to the full suite of techniques from federated learning on compressed communication - any survey on federated learning will list dozens of such techniques. While the question of LoRA-based pre-training is certainly in the zeitgeist given the popularity of LoRA fine-tuning, I don't think that excuses this paper from considering the full set of relevant methods as baselines. Reviewers raised similar concerns and also had some criticisms of the original limited experimental setting considered. The authors did include a good suite of additional experiments in the rebuttal, which should be included in a future revision along with a more comprehensive comparison to preexising methods for large-scale distributed training with limited communication costs.

**Justification For Why Not Higher Score:**

See metareview - the paper does not situate itself with respect to a lot of relevant work.

**Justification For Why Not Lower Score:**

N/A

---

### Decision · Program_Chairs · 2024-01-16

Reject